# Extraterrestrial hexamethylenetetramine in meteorites—a precursor of prebiotic chemistry in the inner solar system

Yasuhiro Oba [1✉], Yoshinori Takano [2], Hiroshi Naraoka [3,4], Yoshihiro Furukawa [5], Daniel P. Glavin [6], Jason P. Dworkin [6] & Shogo Tachibana[7,8]

Despite extensive studies on the formation of organic molecules in various extraterrestrial environments, it still remains under debate when, where, and how such molecules were abiotically formed. A key molecule to solve the problem, hexamethylenetetramine (HMT) has not been confirmed in extraterrestrial materials despite extensive laboratory experimental evidence that it can be produced in interstellar or cometary environments. Here we report the first detection of HMT and functionalized HMT species in the carbonaceous chondrites Murchison, Murray, and Tagish Lake. While the part-per-billion level concentration of HMT in Murchison and Tagish Lake is comparable to other related soluble organic molecules like amino acids, these compounds may have eluded detection in previous studies due to the loss of HMT during the extraction processes. HMT, which can yield important molecules for prebiotic chemistry such as formaldehyde and ammonia upon degradation, is a likely precursor of meteoritic organic compounds of astrochemical and astrophysical interest.

[1] Institute of Low Temperature Science (ILTS), Hokkaido University, N19W8, Kita-ku, Sapporo, Hokkaido 060-0189, Japan. [2] Biogeochemistry Research Center (BGC), Japan Agency for Marine-Earth Science and Technology (JAMSTEC), 2-15 Natsushima, Yokosuka, Kanagawa 237-0061, Japan. [3] Department of Earth and Planetary Sciences, Kyushu University, 744 Motooka, Nishi-ku, Fukuoka, Fukuoka 819-0395, Japan. [4] Research Center for Planetary Trace Organic Compounds (PTOC), Kyushu University, 744 Motooka, Nishi-ku, Fukuoka, Fukuoka 819-0395, Japan. [5] Department of Earth Science, Tohoku University, Sendai 980-8578, Japan. [6] Solar System Exploration Division, National Aeronautics and Space Administration (NASA), Goddard Space Flight Center (GSFC), Greenbelt, MD 20771, USA. [7] UTokyo Organization for Planetary and Space Science (UTOPS), University of Tokyo, 7-3-1 Hongo, Tokyo 113-0033, Japan. [8] Institute of Space and Astronautical Science (ISAS), Japan Aerospace Exploration Agency (JAXA), 3-1-1 Yoshinodai, Sagamihara, Kanagawa 252-5210, Japan. ✉email: oba@lowtem.hokudai.ac.jp

Presence of organic molecules in extraterrestrial environments has been widely accepted thanks to recent successes in the in situ detection of cometary molecules toward 67P/Churyumov-Gerasimenko[1], as well as long-standing astronomical observations[2,3] and analyses of carbonaceous meteorites in laboratories[4–8]. However, despite extensive studies on the formation of organic molecules in various extraterrestrial environments such as molecular clouds[9,10], protosolar nebula[11,12], and asteroids[13–15], it still remains under debate when, where, and how such extraterrestrial molecules were abiotically formed.

A key molecule to solve the problems is hexamethylenetetramine (HMT; $C_6H_{12}N_4$; monoisotopic mass of 140.1062 Da), which is a polyheterocyclic organic molecule (Fig. 1, Supplementary Fig. 1). Based on laboratory experiments simulating photochemical and thermal reactions of interstellar and cometary ice analogues (at ~10 K) initially made of observed molecules, such as water ($H_2O$), ammonia ($NH_3$), and methanol ($CH_3OH$), HMT is in general a significant product (up to 60% by weight) in the total organic products[16–20]. Although the composition of products varies depending on the experimental conditions, HMT is generally abundant especially when methanol is used as an initial reactant[16,18,20]. Since methanol is abundant in interstellar ices[3], the HMT formation is likely to take place in the interstellar medium (ISM) and become incorporated into solar system ices similar to other interstellar molecules[21,22].

Yet HMT has not been observed toward any extraterrestrial environments. Owing to its symmetric tetrahedral structure, HMT does not possess a permanent dipole moment, which precludes its remote observational detection by rotational spectroscopy. Though HMT is an infrared active molecule, its detection in the presence of deep N–H, C–H, and C–N bands in ices, as well as the presence of a strong silicate band at 10 μm, would complicate its definitive identification, so it is also not surprising that it has not yet been observed in interstellar or planetary ices[23]. However, HMT has been postulated to be one of the extended sources of $NH_3$ and HCN in comets[24]. Besides the lack of astronomical detection, there has also been no report on the detection of HMT in any extraterrestrial materials including carbonaceous meteorites, interstellar dust particles, and cometary return samples.

Since HMT is susceptible to degradation by 100 °C water[15] and acid hydrolysis methods[19] traditionally used in meteoritic soluble organic analyses[4]; a different method to extract HMT from meteorites was developed. In the present study, we extracted relatively large portions (masses ranging from 0.5 to 2 g) of interior samples of three carbonaceous chondrites, Murchison, Murray, and Tagish Lake, under mild conditions which utilized neither concentrated acidic solutions nor high temperatures for the extraction processes. The aqueous extracts were purified using cation-exchange chromatography and were then analysed using a high-resolution mass spectrometer (HRMS) coupled with a high-performance liquid chromatograph (HPLC)[19,25]. HMT was successfully detected from Murchison, Tagish Lake, and Murray meteorite extracts at parts-per-billion levels.

## Results

**Detection and quantification of HMT in carbonaceous meteorites.** Figure 2 shows mass chromatograms of the Murchison, Murray, and Tagish Lake meteorites at the mass-to-charge

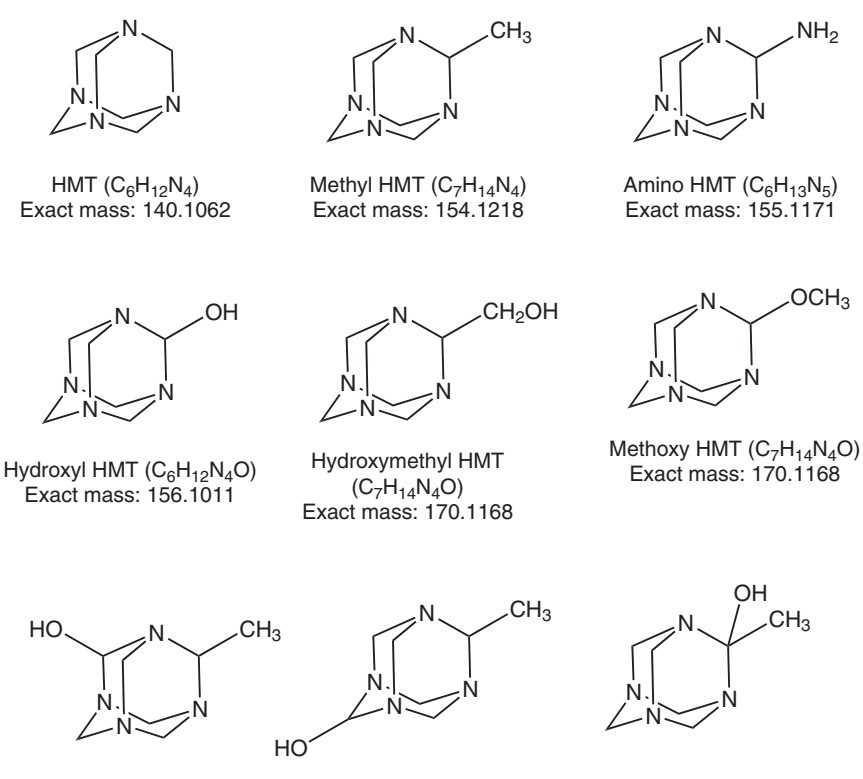

**Fig. 1 Target molecules in the present study.** Molecular structure and exact mass information of hexamethylenetetramine (HMT) and some representative derivatives showing methyl-HMT, amino-HMT, hydroxyl-HMT, hydroxymethyl-HMT, methoxy-HMT, and monohydroxy-monomethyl-HMT discussed in this study. Note that monohydroxy-monomethyl-HMT possesses three structural isomers depending on the positions of the two functional groups.

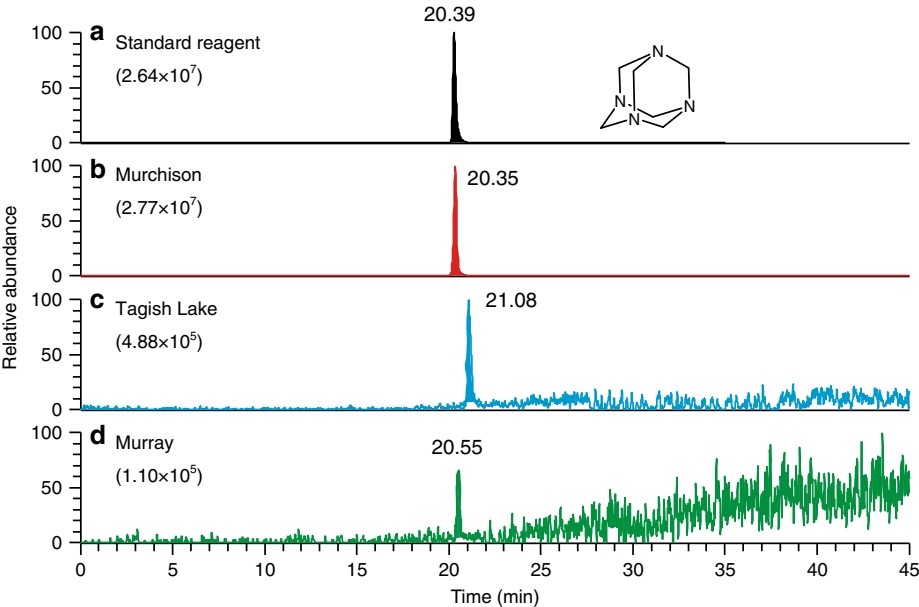

**Fig. 2 Identification of hexamethylenetetramine in meteorites.** Mass chromatograms at the $m/z$ of 141.1135 within a 3 ppm exact mass window at each monoisotopic mass for **a** hexamethylenetetramine (HMT) standard reagent, **b** HMT in Murchison, **c** Tagish Lake, and **d** Murray meteorites, measured using the InertSustain PFP column. The numbers in parenthesis represent the absolute scale in ion intensities for each chromatogram. The numbers near the peak represent the retention time. We note here that a retention time difference between the standard reagent and the target molecule sometimes occurs in a chromatographic separation for complex organic matter[27,47]. To compensate this issue, we always monitored the measured mass within 3-ppm window for the data quality assurance. Also, small levels of fluctuation in the retention time are caused by variations in daily conditions of the liquid chromatograph. The Tagish Lake and Murray extracts were analysed in a different day (the retention time for the HMT standard reagent was 21.07 min) with the Murchison extract.

ratio ($m/z$) of 141.1135 ± 0.0004, which corresponds to the protonated ion of HMT (i.e., $[M + H]^+$ as $[C_6H_{12}N_4 + H]^+$) formed by electrospray ionization (ESI), analysed by a HPLC equipped with an InertSustain PFP analytical column. One sharp peak was observed for each chromatogram at ~20.5 min, which was consistent with HMT standard reagent (Fig. 2a) and far above the blank detection level (Supplementary Fig. 2). The similar consistency was also observed when the sample was analysed under different analytical conditions where Hypercarb or InertSustain Amide was used as a separation column for HPLC analysis (see the "Methods" section and Supplementary Table 1). Based on the retention time and mass accuracy (within 3 ppm of the theoretical $m/z$), even under the different analytical conditions, the observed peak can be confidently assigned to HMT. The observed consistency in the fragmentation pattern of HMT by MS/MS experiments (see the "Methods" section) between the Murchison extract and the standard reagent further supports the above conclusion (Fig. 3). The concentrations of HMT in the three meteorites were 846 ± 37, 29 ± 9, and 671 ± 9 ppb (parts per billion; ng/g meteorite) for Murchison, Murray, and Tagish Lake, respectively (Table 1). Mass peaks attributable to the deuterium (D)-, $^{13}C$-, and $^{15}N$-substituted isotopologues of HMT were also identified in the mass spectra of the Murchison extract (Supplementary Fig. 3). We have confirmed that the loss of HMT is negligible (see the "Methods" section) and that there is no hydrogen isotopic fractionation of HMT during our analytical procedure (Supplementary Fig. 4).

**Tentative detection of HMT-derivatives.** We also observed several peaks with the $m/z$ values well consistent with the HMT derivatives methyl-HMT (HMT-$CH_3$), amino-HMT (HMT-$NH_2$), hydroxy-HMT (HMT-OH), and hydroxymethyl-HMT (HMT-$CH_2OH$), (Fig. 1) in the mass chromatograms at the

$m/z$ of 155.1291, 156.1244, 157.1084, and 171.1240, respectively, as each protonated ion formula in Murchison (Fig. 4). The $m/z =$ 171.1240 trace (Fig. 4e) shows at least three peaks, which might be derived from HMT-$CH_2OH$ and its structural isomers methoxy-HMT (HMT-$OCH_3$) and monohydroxy-monomethyl-HMT (HMT-OH(-$CH_3$)) (Fig. 1). No authentic standards were available, so these assignments are the most likely but other isomers (e.g., ethyl-pentamethylene tetramine instead of HMT-$CH_3$) cannot be excluded. The absence of these species on the mass chromatograms for the HMT standard reagent (Supplementary Fig. 5) indicates that these are likely not formed during workup or clusters or N-functionalizations formed by ESI and so should be indigenous to the meteorite samples. Without authentic standards, an estimate of their possible abundances assumed the same ionization efficiency as HMT; the most abundant derivative was HMT-$CH_3$ (2% of HMT), followed by HMT-$CH_2OH$ or its isomers (<0.6%), HMT-OH (0.2%), and HMT-$NH_2$ (0.03%) (Table 1).

## Discussion

The negligible amounts of HMT in the blank and control samples (see the "Methods" section) compared to the elevated concentrations of HMT measured in the meteorite extracts argue that HMT is indigenous to the meteorites. In addition, the likely detection of several HMT-derivatives also bolsters this conclusion; unlike HMT itself, to our best knowledge, these HMT-derivatives are commercially unavailable and their presence in terrestrial environments has not been reported. However, these HMT-derivatives have been identified in organic residues produced by photolysis of interstellar ice analogues followed by warming to room temperatures, which mimics the processes of molecular evolution toward star formation[17,19]. Furthermore, the estimated relative abundances of these HMT-derivatives in

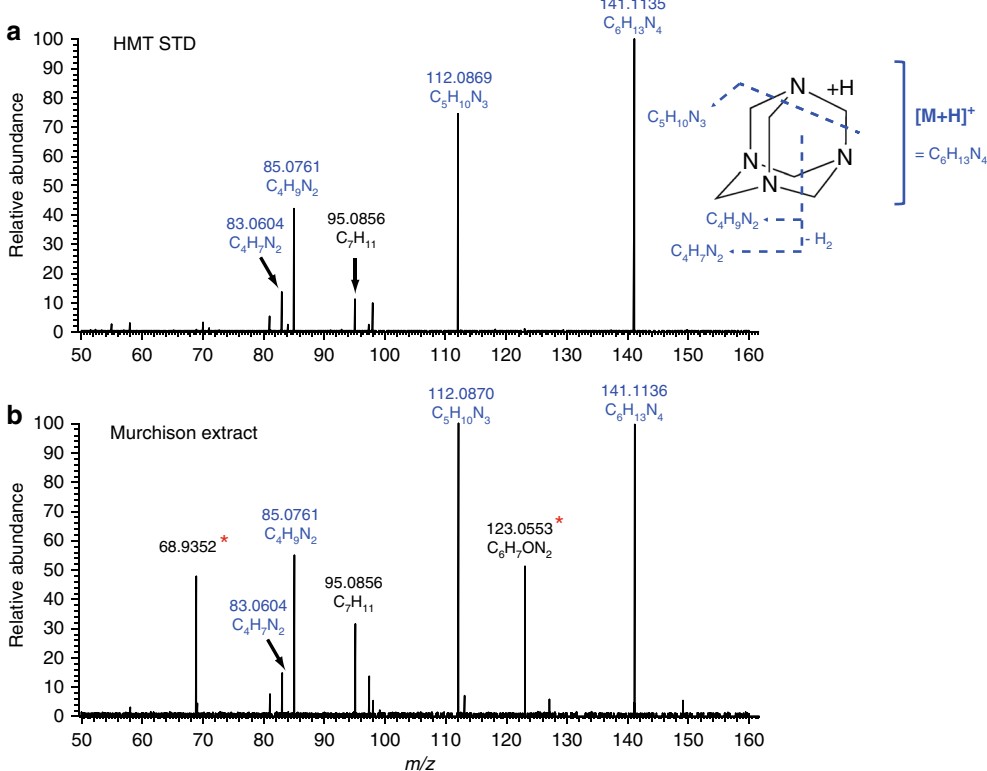

**Fig. 3 Results of MS/MS experiments.** Mass fragmentation patterns of hexamethylenetetramine (HMT) in **a** the standard reagent and **b** the Murchison extract measured by MS/MS experiments (see the "Methods" section). A schematic image of HMT fragmentation is shown alongside the panel **a**. The 6–7 digit numbers in the mass spectra indicate the exact masses of the parent molecule ($C_6H_{13}N_4$: the protonated ion of HMT) and its fragments). The fragmentation patterns are consistent with each other except the presence of peaks with a red asterisk in the Murchison extract, which are derived from other species coexisting with HMT. Note: the mass peak at the $m/z$ of 68.9352 in the Murchison extract could not be successfully assigned to any ions under the assumption that the ion is composed of C, H, N, and O. The mass peak assigned to $C_7H_{11}$ ($m/z = 95.0856$) is a background signal on the LC condition.

**Table 1 Summary of HMT and possible HMT-derivative concentrations and relative abundances.**

| Meteorite | Sample mass extracted (g) | Compound | Formula | Theoretical Mass $M + H^+$ (Da) | Measured Mass $M + H^+$ ($m/z$) | Concentration (ppb)[a] | Relative abundance (%)[b] |
|---|---|---|---|---|---|---|---|
| Murchison | 2 | HMT | $C_6H_{12}N_4$ | 141.1135 | 141.1133 | 846 ± 37 | 100 |
| | | HMT-CH$_3$ | $C_7H_{14}N_4$ | 155.1291 | 155.1290 | 13 ± 0.4 | 2 |
| | | HMT-NH$_2$ | $C_6H_{13}N_5$ | 156.1234 | 156.1235 | 0.3 ± 0.1 | 0.03 |
| | | HMT-OH | $C_6H_{12}N_4O$ | 157.1084 | 157.1081 | 2 ± 0.3 | 0.2 |
| | | HMT-CH$_2$OH and its isomers[c] | $C_7H_{14}N_4O$ | 171.1240 | 171.1237 | <4 ± 0.6 | <0.6 |
| Tagish Lake | 0.5 | HMT | $C_6H_{12}N_4$ | 141.1135 | 141.1134 | 671 ± 9 | 79 |
| Murray | 2 | HMT | $C_6H_{12}N_4$ | 141.1135 | 141.1135 | 29 ± 9 | 3 |

[a]The values represent the average of two measurements with the statistical error.
[b]Relative to HMT in Murchison.
[c]Peaks could not be distinguished between isomers shown in Fig. 1; their upper limit was estimated from the largest peak on the chromatogram.

the organic residues (orders of magnitudes less abundant than HMT)[17,19] are in reasonable agreement with those of the meteoritic HMT-derivatives (Supplementary Fig. 6).

The concentration of HMT in Murchison (846 ± 37 ppb) is within the range of individual water-extractable and acid-produced amino acids (200–5000 ppb)[26] and higher than that of sugars (<180 ppb) and nucleobases (<~70 ppb) in the Murchison meteorite[5,6]. In the Tagish Lake meteorite, the concentration of HMT (671 ± 9 ppb) is also in the range of individual amino acid concentrations identified in acid hydrolysed water extracts of the Tagish Lake meteorite (<14 ppb: Tagish Lake 11i, <1000 ppb: Tagish Lake 11 h)[7]. While in Murray, the concentration of HMT (29 ± 9 ppb) is lower than individual amino acid concentrations (51–2834 ppb) in the same meteorite[8]. It is possible that differences in the Murchison/Tagish Lake and Murray parent body conditions (e.g. temperature, water/rock ratio, etc.) led to lower abundances or higher loss rates of HMT, which may partly be related to the formation of soluble organics. For example, Supplementary Fig. 7 shows plots of the HMT concentrations normalized with glycine concentrations in the same meteorite. There seems no obvious trend in the concentrations of HMT with glycine, suggesting no obvious correlation in terms of their formation history in each meteorite. Supplementary Figs. 8–10 show mass chromatograms of each

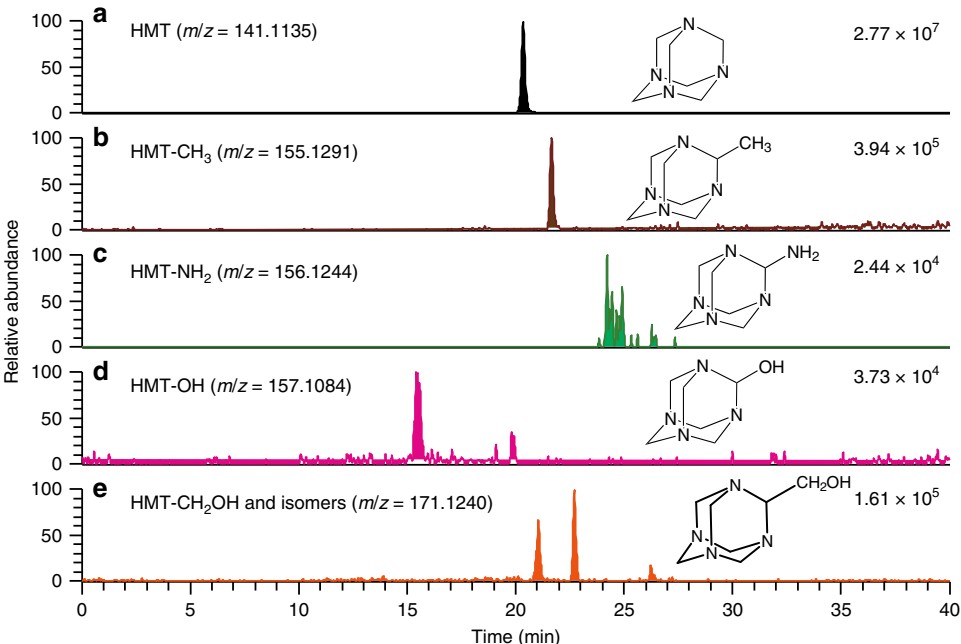

**Fig. 4 Possible identification of hexamethylenetetramine derivatives.** Mass chromatograms at the *m/z* of **a** 141.1135, **b** 155.1291, **c** 156.1244, **d** 157.1084, and **e** 171.1240 (3-ppm window at each monoisotopic mass), which correspond to hexamethylenetetramine (HMT), HMT-CH$_3$, HMT-NH$_2$, HMT-OH, and HMT-CH$_2$OH, respectively, measured using the InertSustain PFP column, in the Murchison meteorite. Mass peaks identified in the panel **e** may include the structural isomers of HMT-CH$_2$OH, such as HMT-OCH$_3$ and HMT-OH(-CH$_3$). The numbers on the upper right in each panel represent the absolute scale in ion intensities for each chromatogram.

meteorite extract at the *m/z* values corresponding to imidazole (C$_3$H$_4$N$_2$; monoisotopic mass of 68.0374 Da) and its alkyl-substituted homologues (up to seven carbon chains), which are proposed as the products after the hydrothermal degradation of HMT[15]. For Murchison and Murray, the presence of alkyl-imidazoles was strongly expected in their extracts; while, they were significantly depleted in Tagish Lake (Supplementary Figs. 8–10). These results do not contradict the assumption that Tagish Lake, at least the specimen used in the present study, could have experienced less extensive hydrothermal alteration than Murchison and Murray on their parent bodies.

Given the harsh extraction conditions of amino acid analyses, one possibility is that some of the HMT and its derivatives can form amino acids during routine amino acid extraction and workup. In fact, acid hydrolysis of HMT-containing organic mixtures yielded amino acids, and the role of HMT for amino acid formation has been investigated well in recent studies[19,27,28]. However, the argument that HMT is the origin of amino acids during workup is weakened by Murray, which has a similar abundance of amino acids to Murchison[8], yet the HMT concentration was lower by about an order of magnitude than Murchison. Moreover, sample heterogeneity between different specimens of the same meteorite, which has been often invoked for explaining different quantitative results of some molecules including their different enantiomeric distributions in the same meteorites[4], can also be invoked. On the other hand, it is likely that HMT is formed during our laboratory workup if both ammonia and formaldehyde are present in the aqueous extract[29]. Previous studies detected both molecules from carbonaceous meteorites after hydrothermal treatment and/or acid hydrolysis of meteorite powders at ~100 °C or above[30–32], implying that both free ammonia and formaldehyde are released from their acid-labile precursors after these treatments. Although it is not clear whether such precursors can contribute to the formation of HMT in aqueous solutions without acid and high-temperature treatment at room temperature, we expect that HMT formed as such

does not constitute a significant fraction in the detected HMT abundance. Nevertheless, there are still a number of uncertainties on the origin of the difference in HMT abundance between three meteorites analysed in the present study (e.g. HMT abundance when each parent body is formed by accretion).

It is reasonable that ISM-derived HMT would be highly D-enriched[19]. Though the typical interstellar values (e.g., D/H ratio ≥0.01)[33] are far higher than seen in any meteoritic compound. No levels of this extreme deuteration of HMT were visible. Yet, it is still possible that the HMT detected has an interstellar provenance and the ISM D was lost to exchange with comparatively D-poor parent body fluids. We have tested the D/H exchange in HMT upon heating with water and silicates to simulate possible variations in the deuteration level of meteoritic HMT through hydrothermal processes in asteroids. When fully deuterated HMT (C$_6$D$_{12}$N$_4$) was heated with H$_2$O under alkaline conditions (pH = 10) at 100 °C, deuterium atoms in HMT were gradually replaced with hydrogen atoms in H$_2$O, resulting in the formation of partly hydrogenated HMT like C$_6$HD$_{11}$N$_4$ and C$_6$H$_2$D$_{10}$N$_4$ after several days (Supplementary Fig. 11). These results suggest that even if HMT was enriched in deuterium upon the formation in the ISM, it might get depleted in deuterium through interactions with relatively deuterium-depleted water on the parent bodies of CM meteorites[34].

Once HMT is incorporated into planetary systems and into a meteorite parent body, it has three likely fates: (1) physico-chemical desorption from the surface of asteroids into the gas phase of the solar system, (2) decomposition, and (3) preservation. It is likely that desorption of HMT from asteroids could be induced either or both by external excitation energies (e.g., cosmic rays and ultraviolet photons) and by thermal processes, although these processes have not been studied experimentally so far. Laboratory studies strongly suggest that aqueous or thermal degradation of HMT on meteorite parent bodies has a potential to yield various kinds of molecules, such as formaldehyde (H$_2$CO), NH$_3$, amines, amino acids, and nitrogen

heterocycles[15,25,35–37], many of which have been identified in carbonaceous meteorites after hydrothermal treatment at around 100 °C or acid hydrolysis[30–32]. HMT that survived these desorption and degradation processes might be delivered to the Earth via meteorites and possibly interplanetary dust particles.

Among the various kinds of molecules which can form via hydrothermal degradation of HMT, both $H_2CO$ and $NH_3$ are considered particularly important for the formation of soluble organic molecules, such as amino acids and sugars, and insoluble organic matter in meteorites through various reactions such as formose and Mannich reactions or Strecker-cyanohydrin synthesis[5,13,15,28,36–40]. Although $H_2CO$ and $NH_3$ are two significant components in interstellar ices[3], which are mainly formed by the hydrogenation of CO and N atoms, respectively[9], due to their low desorption temperatures from interstellar grains (<100 K)[41,42], unless transformed into other (non-volatile) species by chemical reactions, both molecules are likely to be lost from grains during warming up phases toward star formation if the temperature of the grains exceeds the desorption temperature of both molecules. In contrast, since solid HMT does not desorb from grains even at 330 K (refs. [15,18]), it should have more opportunity to be incorporated into inner solar system bodies. Naturally, since HMT is in equilibrium with $H_2CO$ and $NH_3$, it could also have been formed on meteorite parent bodies from both molecules if they are really present, which could keep the HMT concentration relatively constant. However, $H_2CO$ and $NH_3$ have been identified in carbonaceous meteorites upon hydrothermal treatment at around 100 °C or acid hydrolysis[30–32]; conversely these species may be from the decomposition of HMT on the parent body or during laboratory workup. As such, it will be challenging to constrain the location of HMT formation but its presence in the processed interstellar ice analogues[16–20] can be a good indicator to explain its presence in meteorites. Hence, the presence of HMT in carbonaceous meteorites promises its pivotal role to carry interstellar prebiotic precursors to the inner solar system, which should contribute to the chemical evolution in the primordial stage on Earth.

## Methods

**Meteorite samples.** The Murchison meteorite (CM2) was from a 10 g chip taken from a 47.5 g fragment originally from the Field Museum of Natural History, Chicago that had been stored at room temperature in a sealed glass desiccator for many years at the University of Chicago until it was opened in August 2015. The 10 g chip was crushed and homogenized at the NASA Goddard Space Flight Center and a 2 g portion of the powder was sent to Tohoku University. The sample quality (i.e. a degree of contamination) was previously evaluated for amino acids, suggesting very low levels of amino acid contamination based on their heavy carbon isotopic compositions and the detection of racemic alanine[43,44]. The Murray (CM2) and the Tagish lake (C2 ungrouped) meteorites were both from meteorite trading companies with the certification. The exterior surfaces of these meteorite samples were independently washed by 0.1 M HCl solution (water was qTOF grade, Fujifilm Wako Co. Ltd) with a soak (3 min at ambient temperature) and gentle ultra-sonication (0.5 min, <38 kHz by double glass containers) to peel the meteoritic surface layer, and the supernatant was removed. Then, the sample followed an organic solvent soak (3 min at ambient temperature) by dichloromethane/methanol (50:50, v/v) with gentle ultra-sonication (0.5 min, <38 kHz by double glass container). After removing the supernatant, the chemically peeled samples were dried up by a vacuum freeze dryer (EYELA Co., Ltd) at ambient temperature. In a clean bench, the dried samples were gently powdered as fine as using a clean pestle and a clean mortar according to the previous work[5,45] with the present blank test.

**HMT extraction from meteorites and purifications prior to LC analysis.** HMT and other water-extractable hydrophilic molecules (e.g., sugars) were recovered from ~2 g of the Murchison powder and the cation desalting fraction as described in Furukawa et al.[5] was used for this study. For further investigation of other reference carbonaceous meteorites, we conducted the water and solvent extraction for the fine powdered samples (2 g for Murray and 0.5 g of Tagish Lake) using ultra-sonication (10 min with crushed ice in the sonic bath) with two bed-volume of ultra-pure water (qTOF grade, Fujifilm Wako Co. Ltd). After the solid/liquid separation by the centrifugation (10 min, 3000 rpm), the supernatant liquid phase

was recovered; the water-extractable fraction procedures were repeated for three times. The fraction was then frozen and dried up by a vacuum freeze dryer (EYELA Co., Ltd) under ambient temperature. To remove inorganic salts and interfering organic matrix from the extracts, we isolated the HMT fraction using the cation exchange chromatography (AG50W-X8 resin, Bio-Rad Laboratories)[46]. The final elution containing HMTs was dried by a vacuum freeze dryer (EYELA Co., Ltd) under ambient temperature. The final fraction was dissolved in ~1 mL of ultra-pure $H_2O$ and filtered by 0.20 µm PTFE cartridge filter just before the HRMS. The pretreatment eliminates HPLC/ESI-Orbitrap-MS potential artefacts including a chromatographic retention shift[27,47], ion suppression and ion-enhancement effect[48,49]. The recovery of HMT was measured using its standard reagent to be >90%. All glassware and the quartz wool were cleaned by heating in air at 450 °C for 3 hr.

**Identification of HMT by a HRMS coupled with a conventional HPLC.** The meteorite extract was introduced into an Orbitrap mass spectrometer (Q Exactive Plus, Thermo Fischer Scientific) with a mass resolution of $m/\Delta m = $ ~140,000 at a mass-to-charge ratio ($m/z$) of 200 via an HPLC system (UltiMate 3000, Thermo Fischer Scientific) equipped with a reversed-phase separation column (InertSustain PFP, 2.1 × 250 mm, particle size of 3 µm, GL Science) at 40 °C. The eluent programme for this HPLC setup is as follows: solvent A ($H_2O$), solvent B (acetonitrile + 0.1% formic acid by volume) = 90:10 for the initial 5 min, followed by a linear gradient of A:B = 50:50 at 20 min, and it was kept at this ratio for 25 min. The flow rate was 100 µL min$^{-1}$.

The Murchison extract was also analysed using the same HPLC/HRMS equipped with other separation columns: a Hypercarb separation column (2.1 × 150 mm, particle size of 5 µm, Thermo Fischer Scientific)[19] at 30 °C or an InertSustain Amide column (3.0 × 250 mm, particle size of 3 µm, GL Science) at 40 °C in hydrophilic interaction (HILIC) chromatography mode to confirm that the detection of HMT does not depend on analytical columns (Supplementary Table 2). The eluent programme for Hypercarb is as follows: at $t = 0$, solvent A (water), solvent B (acetonitrile + 0.1% formic acid) = 100:0, followed by a linear gradient of A:B = 80:20 at $t = 20$ min and it was kept at this ratio for 5 min. The flow rate was 0.1 mL min$^{-1}$. The eluent programme for the HILIC mode analysis is as follows: at $t = 0$, solvent A (10 mM ammonium formate plus 0.1% formic acid), solvent B (acetonitrile) = 1:99, followed by a linear gradient of A:B = 40:60 at $t = 40$ min and it was kept at this ratio for 5 min. The flow rate was 0.3 mL min$^{-1}$.

The mass spectra were recorded in the positive ESI mode with a $m/z$ range of 50–400 and a spray voltage of 3.5 kV. The capillary temperature of the ion transfer was 300 °C. The injected samples were vaporized at 300 °C. We set up an inverse gradient programme to maintain the ionization efficiency during the ESI. To minimize analytical noise and the background signals in the LC and Orbitrap, we used high purity grade water and acetonitrile (LC/MS grade from Wako Chemical, Ltd.). Under these experimental conditions, the mass precision is always better than 3 ppm for each chromatogram (e.g., 141.1135 ± 0.0004 for protonated HMT).

The MS/MS experiment was also performed using a hybrid quadrupole-Orbitrap mass spectrometer (Q-Exactive Plus, Thermo Fischer Scientific) with the identical HPLC and ionization conditions used for the full-scan analysis. The extracted positive ions $m/z$ 141.11 ± 0.2 were reacted with high-energy (30 in arbitrary unit) collision $N_2$ gas to produce fragmental ions, in which the mass range of $m/z$ 50–160 was monitored by an Orbitrap MS with a mass resolution of ~140,000. The collisions of high-energy $N_2$ with the protonated HMT ion ($m/z$ 141.1135) gave two major fragmental ions; $C_5H_{10}N_3^+$ ($m/z$ 112.0869) and $C_4H_9N_2^+$ ($m/z$ 85.0761), as well as its non-fragmented parent ion ($m/z$ 141.1135, Fig. 3a). The chromatographic peak of the Murchison extract gave the same fragmental ions except for $m/z$ 123.0553 and 68.9352 (Fig. 3b). These mass peaks can be assigned to other species or fragments, which are not related to HMT, coexisting in the Murchison extract. The mass peak assigned to $C_7H_{11}$ ($m/z = $ 95.0856) is a background signal on the LC condition. For the Tagish Lake and Murray meteorites, we were unable to perform MS/MS measurements due to the low concentration of HMT in the extracts.

**Blank test.** The solvent extraction blank analysis with ultra-sonication procedure was performed using 2 g of combusted quartz sand[45] through the same extraction process to verify the potential impurity in the meteorite extracts. The mass chromatogram was shown in Supplementary Fig. 2a. We confirmed that no HMT was identified in this process. In order to evaluate potential terrestrial HMT contamination of the Murchison meteorite from the fall site, the entire wet and dry chemical processes, i.e., the solvent extraction, freeze-drying, desalation, filtering the final fraction, and conditioning of LC-Orbitrap MS, was also applied to the soil sample (102 mg) collected with a clean metal scoop from a depth of 20–30 cm from the Murchison meteorite strewn field in 1999 (please see the supplementary information in the ref. [5]). The mass chromatogram was shown in the Supplementary Fig. 2b. Very tiny amount of HMT was detected in the soil extract with the HMT concentration of 2 ppb, which was <0.5% of the indigenous HMT concentration in the Murchison meteorite extract.

**Deuterium-hydrogen substitution on the hydrothermal treatment of deuterated HMT.** Stock aqueous solution of fully deuterated HMT ($C_6D_{12}N_4$, CDN

Isotopes; HMT-d$_{12}$) was prepared to be 172 mM and pH 10. About 100 μL of the stock solution was transferred to a sample tube (~3 cm in length, 6 mm in diameter) made with pure Au whose one side had been tightly crimped by hand pliers. About 5 mg of amorphous forsterite (~100 nm in diameter) was also enclosed in the same sample tube as an analogue of asteroid minerals. After the headspace of the tube was purged with dry N$_2$ gas, the other side was also crimped, and the sample tube was heated at 100 °C for up to 31 days using an autoclave (MMS-50, OM Labotec, Japan). We confirmed the weight of the sample tube did not change after heating, which indicated effectively no sample loss from the tube. The heated HMT solution was extracted from the sample tube, filtered by a hydrophilic PTFE membrane filter (Millex®-LH 0.45 μm, Merck Millipore) to remove the silicate powders, and analysed by HRMS using a Thermo Scientific Exactive by flow injection. Analytical details have been reported in Oba et al.[50]

## Data availability
The data that support the findings of this study are available from the corresponding author (Y.O.) upon reasonable request.

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

## Acknowledgements

This work is dedicated to the late Prof. Akira Shimoyama, a pioneer of organic cosmochemistry in meteorites. We thank Prof. Akira Tsuchiyama (Ritsumeikan University) for providing amorphous forsterite powder, Dr. Minako Hashiguchi (Nagoya University) for her technical advice on the sample analysis by the Orbitrap MS. Prof. Akira Kouchi and Prof. Naoki Watanabe (ILTS, Hokkaido University) are acknowledged for the discussion on the formation of HMT in interstellar environments. We also thank Dr. Robert Minard and the Dr. Clifford N. Matthews' research group at the University of Chicago for providing the Murchison meteorite and Professor Reid R. Keays from the University of Melbourne for collecting and providing the Murchison soil sample. This work was partly supported by JSPS KAKENHI Grant Numbers JP15H05749, JP16H04083, JP17H04862, and JP20H00202 as well as NASA Astrobiology Institute through award 13-13NAI7-0032 to the Goddard Center for Astrobiology, NASA's Planetary Science Division Internal Scientist Funding Programme through the Fundamental Laboratory Research (FLaRe) work package at NASA Goddard Space Flight Center, and a grant from the Simons Foundation (SCOL award 302497 to J.P.D.).

## Author contributions

Y.O. and Y.T. designed this project in consultation with H.N. and S.T. For sample preparation, D.P.G. and J.P.D. performed the pretreatment of the Murchison sample with the quality assessment. Y.T. and Y.F. extracted HMT from meteorites and purified it before HPLC–MS analysis. Y.T., Y.F. and D.P.G. conducted an assessment using a reference soil sample from the Murchison meteorite fall locality in Murchison, Australia. Y.O., Y.T., and H.N. analysed the sample. D.P.G. and J.P.D. refined the knowledge of extractable meteoritic organic molecules. Y.O., Y.T., and H.N. wrote the paper. All authors commented on the manuscript.

## Competing interests

The authors declare no competing interests.
