## [Peer Review File · Nature Communications]

REVIEWER COMMENTS

Reviewer #1 (Remarks to the Author):

The authors show the presence of hexamethylenetetramine (HMT) in carbonaceous chondrites. It is potentially highly significant results, since HMT is a compound with interests in astrochemistry and astrobiology points of view, but never detected in meteorites, comets, and ISM. If HMT and its derivatives are really extraterrestrial, it is worth to publish in Nature Communications. However, I have important concerns.

First, HMT is easy to decompose, but in addition, it is very easy to be formed from formaldehyde and ammonia at room temperature, particularly in acidic conditions (Vinogradoff et al., *Phys. Chem. Chem. Phys.*, 2012, 14, 12309–12320, and references there in). Thus, HMT could be formed during extraction procedures since formaldehyde and ammonia exist in these meteorites. It is consistent with the blank analyses which did not contain these precursors of HMT. Please also indicate the concentrations of formaldehyde and ammonia in these meteorites (from literatures).

Even if the HMT is not formed during extraction procedures, the interstellar origin of the meteoritic HMT is doubtful considering the fragile nature of the HMT. How could HMT survive millions of years of aqueous alteration and billions of years of preservation in the parent body(ies)?

In lines 119-122, the authors imply that the Tagish Lake specimen used in their study may have experienced thermal alteration less extensive than Murchison and Murray, due to the less abundance of alkyl-imidazoles which are likely produced from HMT. It is too speculative. There are way too many compounds in such chondrites (e.g., Schmitt-Kopplin et al., 2010, PNAS doi: 10.1073/pnas.0912157107) and direct relation in abundances of HMT and alkyl-imidazoles is unrealistic.

In lines 160-164, the authors mention that formaldehyde and ammonia could not be incorporated into bodies inside the water snow line, but parent bodies of carbonaceous chondrites could be formed far from the current asteroid belt, as proposed by such as Nice model.

Reviewer #2 (Remarks to the Author):

Dear Authors,

I was very pleased to receive this review because, if confirmed, such results are very important for the astrochemical community and will rekindle the debate about the origins of the organic matter in meteorites. The MS is well written and easy to read, data are well presented and discussion is interesting and well documented.

However, there are 3 major issues with the data that has to be done to firmly confirm the detection of HMT and derivatives in the soluble organic matter of meteorite.

1/ While the authors present chromatograms with retention time and extracted mass (m/z 141.1135) for HMT in the meteorites and as standard, it has been shown by several studies now that it is not sufficient to confirm the presence of molecule in samples with a high molecular diversity, such as the soluble organic matter of meteorite. The fragmentation pattern of the peak is request. The presence of HMT in ice analog residues has been for example confirmed by fractionation pattern (in addition to extracted mass) (Danger et al., *GCA*, 118 (2013) 184-201, <https://doi.org/10.1016/j.gca.2013.05.015>). But the search for HMT in Tholin samples (titan analogues), by HPLC-Orbitrap was unsuccessful because of the fractionation pattern, which invalidate the attribution of the peak detected by Orbitrap at 141.1131 to HMT (Gautier et al., *Icarus*, 275 (2016) 259–266, <https://doi.org/10.1016/j.icarus.2016.03.007>). Multiple isomers (20 to 30!) are possible for one peak detected by high mass spectrometry for the soluble organic matter of meteorites (Ruf et al., *Life*, 9 (2019) <https://www.mdpi.com/2075-1729/9/2/35>; Schmitt-kopplin et al, PNAS, 107 (2010) 2763-2768, <https://doi.org/10.1073/pnas.0912157107>), so the maximum of prudence is request to validate the presence of a molecule under one mass even if UPLC are used to add a 2nd dimension.

For me your results validate only two third of the requested proofs to confirm the detection of HMT. For

example, and I am sure the authors are aware of these studies, Oba et al., (Nature, 10 (2019) <https://www.nature.com/articles/s41467-019-12404-1>) have previously reported the detection of 3 pyrimidine nucleobases using the same analytical procedure as here, and few months' later, another group shows that among the nucleobases detected by Oba et al, only one was validated with the additional method using the fragmentation pattern on the peak (Ruf et al., ApJL, Vol 887, (2019) <https://iopscience.iop.org/article/10.3847/2041-8213/ab59df>). The other were isomers. Of course it requests more analytical time and data treatment. Nonetheless, all these papers paper cited above highlight the difficulty to identify with a maximum certainty a molecule in complex mixtures, even using high technology such as UPLC Orbitrap-MS.

My recommendation would be to perform this additional analysis, MS-MS fractionation of the peak at 141.1135 in parallel to HMT standard. Your mass spectro (Q exactive plus) should be able to do that, so please confirm your result because it is important for the community.

2/ My second point concern the HMT-derivatives molecules. Here again the fractionation is requested to propose their presence in the soluble organic matter of meteorites, also because no standards can be found for such compounds. So for these compounds you have only one third of the requested proofs to validate their presence. Please show coherent fragmentation pattern with the structure, to propose these molecules, as this has been done in Danger at al., <https://doi.org/10.1016/j.gca.2013.05.015>). Right now this is only speculation.

3/ My third major point concern the reaction yield of your protocol. Could you perform a blank sample with HMT (and not only with sand), submitted to the entire same process as the meteorite sample, to determine the possible alteration of HMT and the reaction yield of your measurement. Is your quantification over-estimate? Under-estimate? My concern is also regarding the formation of HMT by formaldehyde and ammonia directly during your extraction protocol (please see Kebukawa et al, 2019, <https://doi.org/10.1016/j.icarus.2020.113827>). Could you comment on that in the MS ?

After the completion of these major concerns, I would be very pleased to consider the paper for publication and review details.

Reviewer #3 (Remarks to the Author):

I have read the paper entitled "Hexamethylenetetramine in meteorites: a precursor of volatiles in the inner solar system", by Y. Oba et al., which reports the detection of HMT in three carbonaceous chondrites (Murchison, Murray, and Tagish Lake). The detection of HMT and its derivatives seems robust, with the caveats associated with identifications being clearly exposed. The detection of HMT in meteorites is a crucial step towards being able to link laboratory astrochemistry (which has predicted the existence of HMT in meteorites for a long time, most likely inherited from the ISM) and the analysis of asteroidal samples and their connection to the formation of the Earth and early life. Many astrochemists have previously tried to identify HMT in meteorites and I congratulate the authors for having been able to make such good progress on this topic. That said, the discussion of the paper remains a bit superficial, and I find it a shame that the paper turned out to look like a "data report". I think your contribution would be stronger with more general background about prebiotic chemistry and the link between organic molecules present in the ISM, in chondrites and potentially on early Earth. I would recommend introducing the idea that multiple pathways of extraterrestrial organosynthesis have been proposed in the literature (e.g., in the ISM (Nuevo 2011 or some other reference), in the gas phase of the PSN (Bekaert et al. 2018 APJ), and/or on asteroids (e.g., Kebukawa et al. 2017)). It would also be great to emphasize the implications of finding or not HMT in meteorites, and why it is important. I would also suggest adding some background about previous attempts to find HMT in meteorites. Lastly, I find the term "volatiles" in the title very misleading, as your paper does not deal with the origin of water in the inner Solar System for instance. I strongly suggest changing "volatiles" for "prebiotic chemistry".

In summary, more discussion about the relevance of this finding, the connection between ISM processes and

molecules found in chondrites, and the products of HMT degradation (e.g., to form amino acids), is needed before I would consider this paper suitable for publication in Nature Communications. Additional comments are reported here below.

David V. Bekaert

additional comments/suggestions :

I believe "Rubin, M., Bekaert, D. V., Broadley, M. W., Drozdovskaya, M. N., & Wampfler, S. F. (2019). Volatile species in comet 67P/Churyumov-Gerasimenko: investigating the link from the ISM to the terrestrial planets. ACS earth and space chemistry, 3(9), 1792-1811." would be a good reference to cite in order to introduce the concept of the ISM-comet-Earth connection

I would delete "likely" L37

I would change "volatile" for "prebiotic" L38

I would add "at" before "~10K" L43

Please add a reference to the sentence L45-47

Please reformulate "its detection in the presence of deep N-H, C-H, and C-N bands in ices would complicate its definitive identification". I guess what you want to say is that the presence of deep N-H, C-H, and C-N bands in ices would prevent straightforward identification of HMT

Maybe remove the "s" from "detections" L56

I would remove "for the extraction processes to search for HMT" L64

L73-76: can you show the spectra as well?

L81: Why did you not identify these as well for the two other meteorites?

L103-105: can you please consider making a figure to show how reasonable this agreement is? (and add this to the supplements?)

L106: delete "the" or add "aliquot" after Murchison

L109: change "was" for "is" to be consistent with the previous sentence

"lower abundances or higher loss rates" L112 can you precise whether you are talking about HMT, amino acids, both?

L114-115: how the absence of an obvious trend can support a weak correlation in terms of their formation history? I suggest putting "no obvious correlation" or add more details if you want to suggest there is some correlation despite the absence of evidence for it

Maybe consider changing "processes" by "degradation" L119

L120: this sentence does not need a ";"

was it "expected" or "observed"? please be specific

L122: I am confused. you say L119 that these molecules are "proposed as the products after 119 the hydrothermal processes of HMT" and then use their abundances to talk about the extent of "thermal alteration" L122. It is important to distinguish hydrothermal (i.e., aqueous) alteration and thermal alteration (which does not necessarily requires aqueous conditions). Please be specific

L119-122: can you refer to a figure/reference?

L123-124: this is a personal opinion, but I really don't like presenting a hypothesis as a question. I would suggest saying "one possibility is that" or something along these lines

L126-128: there are a couple of studies that have investigated the potential link between HMT and amino acids, and I am aware that this one is for instance coming out

V. Vinogradoff, L. Remusat, H. McLain, J. Aponte, S. Bernard, G. Danger, J. Dworkin, J. Elsila, M. Jaber (2020) "Impact of phyllosilicates on amino acid formation under asteroidal conditions". ACS Earth and Space Chemistry. In press

Maybe you want to add more reference about this

L129: remove "naturally"

L131: add a coma after meteorites

L134-135: please merge the two sentences

"analyzed" L380 – please check that you are consistently being using english vs. american

L140: asteroidS

L141: at 100°C is it water vapor? What was the pressure of your experiment?

L148: "gas phase of the solar system" are you exclusively talking about the protoplanetary disk period of

time? Or if this happens today, it would be desorption to vacuum

L154: you can also add reference to Vinogradoff's work here

L155: "dust" are you specifically talking about interplanetary dust particles? Please be specific

L163-164: can you please add some information about the past location of the snow line during the evolution of the solar system. The location of the snowline in the protoplanetary disk is a direct function of the stellar accretion rate and is expected to move inward as the accretion rate decreases with time (Bitsch et al., 2015). Based on astronomical observations, stellar accretion rates are estimated to have decreased on average from $10^{-8} M_{\odot}/y$ at 1 My to $1-5 \cdot 10^{-9} M_{\odot}/y$ at 3 My (M_{\odot} = the solar mass) (Hartmann et al., 1998), inducing the snowline to drift from 3 AU to 1 AU (Bitsch et al., 2015). Such a snowline drift implies that water ice grains were ubiquitous in the inner Solar System, and formed abundant ice-rich planetesimals. I'm not asking you to add all of this stuff, but at least add provide the reader with some intuition regarding the spatial evolution of the snow line during the evolution of the PPD

L169: what do you mean by workup? lab processing?

L171: I suggest changing "volatiles" for "prebiotic precursors" and add "inner" before "solar system"

L179-180: "The sample quality was previously evaluated for amino acids" what does it mean? It just means you analyzed amino acids? How does it relate to the sample quality?

L256: space missing after 1999

I would remove the molecular structure from Fig. 2 as it looks like it only concerns the first spectrum.

Replies to comments by Reviewers

We appreciate the constructive review comments from three Reviewers on our manuscript (NCOMMS-20-26492-T) entitled “Hexamethylenetetramine in meteorites: a precursor of volatiles in the inner solar system”. We carefully read the whole comments and modified the original version of the manuscript based on their helpful comments. The changes we made based on the Reviewer’s comments are noted in red font in the revised manuscript/supporting information. Our replies to each comment (Times New Roman) are denoted below following to the reviewer’s comments (Arial). Note that in our replies below, we denoted Line numbers in the revised manuscript.

Reviewer #1 (Remarks to the Author):

The authors show the presence of hexamethylenetetramine (HMT) in carbonaceous chondrites. It is potentially highly significant results, since HMT is a compound with interests in astrochemistry and astrobiology points of view, but never detected in meteorites, comets, and ISM. If HMT and its derivatives are really extraterrestrial, it is worth to publish in Nature Communications. However, I have important concerns.

[Reply] Thank you very much for your constructive comments. We carefully read your comments and replied to all of them as follows:

[Comment 1] First, HMT is easy to decompose, but in addition, it is very easy to be formed from formaldehyde and ammonia at room temperature, particularly in acidic conditions (Vinogradoff et al., Phys. Chem. Chem. Phys., 2012, 14, 12309–12320, and references there in). Thus, HMT could be formed during extraction procedures since formaldehyde and ammonia exist in these meteorites. It is consistent with the blank analyses which did not contain these precursors of HMT. Please also indicate the concentrations of formaldehyde and ammonia in these meteorites (from literatures).

[Reply] We understand that HMT is very easy to be formed from free formaldehyde and ammonia, even at room temperature. However, we consider that the formation of HMT during the meteorite extraction procedure is questionable. Reviewer 1 pointed out the presence of both formaldehyde and ammonia in these meteorites probably based on literature reports (e.g. Pizzarello et al. 1994, GCA, Pizzarello and Holmes 2009, GCA). The concentration of NH₃ in the Murchison is reported to be ~0.3 μmol/g (Pizzarello et al.

1994; Pizzarello and Holmes 2009) (i.e. ~ 6 nmol/g (846 ppb) for HMT in Murchison, Table 1). It should be noted that NH_3 was extracted from Murchison after strong acid hydrolysis at temperatures above 100°C in these previous studies. Pizzarello and coworkers (Pizzarello et al. 2011, PNAS, Pizzarello and Williams 2012, ApJ) also reported the detection of ammonia after the hydrothermal treatment at 300°C and 100 MPa for 6 days of insoluble organic matter (IOM) in the Murchison and Tagish Lake meteorites. However, to our knowledge, there is no published reports of the detection of “labile” ammonia which can be extractable even under the mild (i.e. no acid, near room T) conditions, strongly implying that there is little or no labile ammonia in these carbonaceous meteorites. Instead, there should be precursor(s) of NH_3 which can yield NH_3 under acidic conditions at $\sim 100^\circ\text{C}$. One of the possible precursors might be some ammonium salts as have been discovered in comet 67P and potentially some asteroids (Poch et al. 2020 Science Vol. 367, Issue 6483, eaaw7462; Altwegg et al. 2020 Nat. Astron. Vol. 4, 533–540), and others would include HMT since it releases NH_3 upon acid hydrolysis. Unlike “free” ammonia, it is not clear whether such ammonium salts can be used for HMT formation not only on meteorite parent bodies but also during analytical procedures in the laboratory. Hence, we believe that the formation of HMT on the parent bodies and during extraction procedures is still nontrivial. According to Pizzarello and Holmes (2009), the concentration of formaldehyde, which was extracted at $80\text{--}100^\circ\text{C}$ for 42 hours in total, is 10 nmol/g in Murchison. In Tagish Lake, the concentration of formaldehyde extracted in a similar way is 3 to 22 nmol/g (Simkus et al. 2019). Accordingly, we added the following red sentence into Lines 164-166: “..., many of which have been identified in carbonaceous meteorites **after hydrothermal treatment at around 100°C or acid hydrolysis**”.

Also, we modified the following sentence into Lines 178-180 as follows: “Naturally, since HMT is in equilibrium with H_2CO and NH_3 , it could also have been formed on meteorite parent bodies **from both molecules, although they have been identified in carbonaceous meteorites upon hydrothermal treatment at around 100°C or acid hydrolysis**; conversely...”

[Comment 2] Even if the HMT is not formed during extraction procedures, the interstellar origin of the meteoritic HMT is doubtful considering the fragile nature of the HMT. How could HMT survive millions of years of aqueous alteration and billions of years of preservation in the parent body(ies)?

[Reply] Although HMT is fragile in aqueous solution, in particular at low pH, it would

not be as fragile in the absence of liquid water or acidic solutions at moderate temperatures (e.g. Blazevic et al. 1979). So, the presence of HMT in meteorites suggests that meteoritic HMT was not exposed to such aqueous solutions for long durations enough to be decomposed in parent bodies.

[Comment 3] In lines 119-122, the authors imply that the Tagish Lake specimen used in their study may have experienced thermal alteration less extensive than Murchison and Murray, due to the less abundance of alkyl-imidazoles which are likely produced from HMT. It is too speculative. There are way too many compounds in such chondrites (e.g., Schmitt-Kopplin et al., 2010, PNAS doi: 10.1073/pnas.0912157107) and direct relation in abundances of HMT and alkyl-imidazoles is unrealistic.

[Reply] We rephrased the sentence in the lines 120-122 as follows: “**These results do not contradict with an assumption that Tagish Lake, at least the specimen used in the present study, could have experienced hydrothermal alteration less extensive than Murchison and Murray on their parent bodies.**”

[Comment 4] In lines 160-164, the authors mention that formaldehyde and ammonia could not be incorporated into bodies inside the water snow line, but parent bodies of carbonaceous chondrites could be formed far from the current asteroid belt, as proposed by such as Nice model.

[Reply] Even if parent bodies of carbonaceous chondrites were formed far from the current asteroid belt, as long as the position of the “ammonia (and formaldehyde) snow line” is unknown, we cannot reach any decisive conclusion on this point. So, to avoid any discussion about such unanswered question, we deleted the sentence “This is in particular ... where the asteroid belt is located.” Instead, we added the following sentence into Line 175: “...toward star formation **if the temperature of the formed asteroid exceeds the desorption temperature of both molecules**”.

Reviewer #2 (Remarks to the Author):

[Comment 1]

Dear Authors,

I was very pleased to receive this review because, if confirmed, such results are very important for the astrochemical community and will rekindle the debate about the origins of the organic matter in meteorites. The MS is well written and easy to read, data are well presented and discussion is interesting and well documented.

However, there are 3 major issues with the data that has to be done to firmly confirm the detection of HMT and derivatives in the soluble organic matter of meteorite.

[Reply] Thank you very much for your very positive comments. We carefully read the comments and modified the manuscript as shown below.

[Comment 1] While the authors present chromatograms with retention time and extracted mass (m/z 141.1135) for HMT in the meteorites and as standard, it has been shown by several studies now that it is not sufficient to confirm the presence of molecule in samples with a high molecular diversity, such as the soluble organic matter of meteorite. The fragmentation pattern of the peak is request. The presence of HMT in ice analog residues has been for example confirmed by fractionation pattern (in addition to extracted mass) (Danger at al., GCA, 118 (2013) 184-201, <https://doi.org/10.1016/j.gca.2013.05.015>). But the search for HMT in Tholin samples (titan analogues), by HPLC-Orbitrap was unsuccessful because of the fractionation pattern, which invalidate the attribution of the peak detected by Orbitrap at 141.1131 to HMT (Gautier et al., Icarus, 275 (2016) 259–266, <https://doi.org/10.1016/j.icarus.2016.03.007>). Multiple isomers (20 to 30!) are possible for one peak detected by high mass spectrometry for the soluble organic matter of meteorites (Ruf et al., Life, 9 (2019) <https://www.mdpi.com/2075-1729/9/2/35>; Schmitt-kopplin et al, PNAS, 107 (2010) 2763-2768, <https://doi.org/10.1073/pnas.0912157107>), so the maximum of prudence is request to validate the presence of a molecule under one mass even if UPLC are used to add a 2nd dimension. For me your results validate only two third of the requested proofs to confirm the detection of HMT. For example, and I am sure the authors are aware of these studies, Oba et al., (Nature, 10 (2019) <https://www.nature.com/articles/s41467-019-12404-1>) have previously reported the detection of 3 pyrimidine nucleobases using the same analytical procedure as here, and

few months' later, another group shows that among the nucleobases detected by Oba et al, only one was validated with the additional method using the fragmentation pattern on the peak (Ruf et al., ApJL, Vol 887, (2019) <https://iopscience.iop.org/article/10.3847/2041-8213/ab59df>). The other were isomers. Of course it requests more analytical time and data treatment. Nonetheless, all these papers paper cited above highlight the difficulty to identify with a maximum certainty a molecule in complex mixtures, even using high technology such as UPLC Orbitrap-MS. My recommendation would be to perform this additional analysis, MS-MS fractionation of the peak at 141.1135 in parallel to HMT standard. Your mass spectro (Q exactive plus) should be able to do that, so please confirm your result because it is important for the community.

[Reply] We do not think that MS/MS analyses are necessary to firmly assign HMT in our samples due to the following reasons. First, the retention time of the detected HMT in meteorites was very close (< 0.04 min for the analysis by PFP column) to the standard reagent analysed under the same conditions. As described in the original manuscript, this consistency has also been confirmed using different analytical columns (Hypercarb and InterSustain Amide). In addition, the measured mass of the HMT in the form of its protonated ion (141.1133-141.1135) was also very close to the theoretical mass (141.1135). These consistencies confirmed by different analytical methods guarantee that the detected peak is surely derived from HMT itself. Since Danger et al. (2013) did not use HPLC in their analysis, other techniques such as MS/MS analysis could be helpful to identify a target molecule in a mixture of complex species. Comparison with HMT in the tholin samples by Gautier et al. (2016) is meaningless since there is no peak whose retention time is consistent with that of HMT standard reagent (Figure SI 6 in Gautier et al. 2016), which implies that HMT is not present in their samples. In the case of nucleobases in organic residues produced by photochemical reactions of interstellar ice analogues, since multiple structural isomers are observed very close to target bases, MS/MS analysis could be helpful to identify them. However, in the meteorite extracts at the m/z of 141.1135 (within 3 ppm), only one peak (except noise) appeared on each mass chromatogram (Figure 2), a result that does not require follow-up MS/MS analysis. Thus, we would like to add a Supplementary Table 1 which includes comparisons of retention times (normalized as Δt) and the exact mass accuracy (normalized as $\Delta m/z$) under the three different analytical conditions, supporting the precise HMT identification. Also, the detailed information about the three separation columns was summarized in Supplementary Table 2.

[Comment 2] My second point concern the HMT-derivatives molecules. Here again the fractionation is requested to propose their presence in the soluble organic matter of meteorites, also because no standards can be found for such compounds. So for these compounds you have only one third of the requested proofs to validate their presence. Please show coherent fragmentation pattern with the structure, to propose these molecules, as this has been done in Danger at al., <https://doi.org/10.1016/j.gca.2013.05.015>). Right now this is only speculation.

[Reply] As already mentioned in the manuscript and as the Reviewer 2 has already understood, HMT-derivatives are not commercially available. Of course, we can analyse the sample using MS/MS. However, as long as the obtained results cannot be compared with those from their standard reagents, the derived conclusion is no more than speculation. The main purpose of the present study is to show the presence of HMT (NOT HMT-derivatives). Therefore, we do not further analyse the meteorite extract using MS/MS here. However, if standard reagents of HMT-derivatives become available in future, we will readily try to identify such species in the meteorite extracts.

[Comment 3] My third major point concern the reaction yield of your protocol. Could you perform a blank sample with HMT (and not only with sand), submitted to the entire same process as the meteorite sample, to determine the possible alteration of HMT and the reaction yield of your measurement. Is your quantification over-estimate? Under-estimate ? my concern is also regarding the formation of HMT by formaldehyde and ammonia directly during your extraction protocol (please see Kebukawa et al, 2019, <https://doi.org/10.1016/j.icarus.2020.113827>). Could you comment on that in the MS ?

After the completion of these major concerns, I would be very pleased to consider the paper for publication and review details.

[Reply] We have compared both the absolute abundance of HMT and the relative abundances of deuterated HMT isotopologues formed by photochemical processes of interstellar ice analogues containing deuterated methanol (e.g. Oba et al. 2017, ApJ, 849, 122) before and after the present extraction and purification processes. We found no significant loss of HMT. In addition, the relative abundances of the deuterated isotopologues did not change after these processes (see Figure A below). These results mean that the present analytical procedure does not change the molecular and isotopic

composition of HMT in the sample. We would like to add Figure A into the revised manuscript as a Supplementary Figure 3. Also, the following sentence was added to Lines 82-84: “We have confirmed that the loss of HMT is negligible and that there is no hydrogen isotopic fractionation of HMT during our analytical procedure (Supplementary Fig. 3)”.

Regarding the formation of HMT during our extraction protocol, if labile ammonia and formaldehyde are present in the aqueous extract, it is very likely that HMT is synthesized in the solution, as proposed by Kebukawa et al. (2020). However, it is not evident whether this is true in the present case. In previous studies on the detection of ammonia and formaldehyde from meteorites (e.g. Pizzarello et al. 1994; Simkus et al. 2019), both molecules were extracted by hydrothermal treatment at ~100 °C and/or after acid hydrolysis at 100 °C. In other words, these molecules would not have been detected by gentle extraction procedures (i.e. moderate T, no acid) from meteorites. These imply that both molecules are not present in their “free” form (i.e. NH₃ and H₂CO), but rather their precursors are present in meteorites, which would release formaldehyde and ammonia upon such treatments. Although it is not clear whether such precursors can contribute to the formation of HMT without any hydrothermal treatment and acid hydrolysis, we strongly expect that this is not the case. Hence, we do not think the HMT formation through our much lower temperature extraction protocol would have resulted in a significant contribution to the detected HMT. Thus, we added the following sentences into the revised manuscript (note: references in the following text are appropriately numbered in the revised manuscript): “On the other hand, it is likely that HMT is formed during our laboratory workup if both ammonia and formaldehyde are present in the aqueous extract (Kebukawa et al. 2020). Previous studies detected both molecules from carbonaceous meteorites after hydrothermal treatment and/or acid hydrolysis of meteorite powders at ~100°C or above (Pizzarello et al. 1994; Pizzarello and Holmes 2009; Simkus et al. 2019), implying that both free ammonia and formaldehyde are released from their acid-labile precursors after these treatments. Although it is not clear whether such precursors can contribute to the formation of HMT in aqueous solutions without acid and high-temperature treatment at room temperature, we expect that HMT formed as such does not constitute a significant fraction in the detected HMT abundance.”

Figure A. Relative abundances of deuterated HMT isotopologues (d_n , where n is the number of D atoms in an HMT isotopologue) before and after the extraction and purification procedures. The dashed red line represents a 1:1 correlation.

Reviewer #3 (Remarks to the Author):

[Main comment] I have read the paper entitled "Hexamethylenetetramine in meteorites: a precursor of volatiles in the inner solar system", by Y. Oba et al., which reports the detection of HMT in three carbonaceous chondrites (Murchison, Murray, and Tagish Lake). The detection of HMT and its derivatives seems robust, with the caveats associated with identifications being clearly exposed. The detection of HMT in meteorites is a crucial step towards being able to link laboratory astrochemistry (which has predicted the existence of HMT in meteorites for a long time, most likely inherited from the ISM) and the analysis of asteroidal samples and their connection to the formation of the Earth and early life. Many astrochemists have previously tried to identify HMT in meteorites and I congratulate the authors for having been able to make such good progress on this topic. That said, the discussion of the paper remains a bit superficial, and I find it a shame that the paper turned out to look like a "data report". I think your contribution would be stronger with more general background about prebiotic chemistry and the link between organic molecules present in the ISM, in chondrites and potentially on early Earth. I would recommend introducing the idea that multiple pathways of extraterrestrial organosynthesis have been proposed in the literature (e.g., in the ISM (Nuevo 2011 or some other reference), in the gas phase of the PSN (Bekaert et al. 2018 APJ), and/or on asteroids (e.g., Kebukawa et al. 2017). It would also be great to emphasize the implications of finding or not HMT in meteorites, and why it is important. I would also suggest adding some background about previous attempts to find HMT in meteorites. Lastly, I find the term "volatiles" in the title very misleading, as your paper does not deal with the origin of water in the inner Solar System for instance. I strongly suggest changing "volatiles" for "prebiotic chemistry".

In summary, more discussion about the relevance of this finding, the connection between ISM processes and molecules found in chondrites, and the products of HMT degradation (e.g., to form amino acids), is needed before I would consider this paper suitable for publication in Nature Communications. Additional comments are reported here below.

David V. Bekaert

[Reply] We appreciate the detailed comments by the Reviewer 3, Dr. Bekaert. We carefully read them and modified the manuscript.

Firstly, we added a sentence to cite previous studies on the formation of organic molecules in various extraterrestrial environments into Lines 29-31 as follows: “However, despite extensive studies on the formation of organic molecules in various extraterrestrial environments such as molecular clouds (Hama & Watanabe 2013, Chem. Rev., Oberg 2016, Chem. Rev.), protosolar nebula (Bekaert et al. 2018 ApJ, Gautier et al. 2020, EPSL), and asteroids (Kebukawa et al. 2017, Sci. Adv.; Nakano et al. 2020, Sci. Rep.), ...” (Note: the references cited here are appropriately numbered in the revised manuscript).

Secondly, we have already noted the importance of HMT in meteorites in the final paragraph of the main text.

Thirdly, we unfortunately could not find previous studies which tried to find HMT in meteorites. We have already mentioned the non-detection of HMT so far in extraterrestrial materials in the initial paragraph. Moreover, we prefer not to mention previous studies which were not successful to detect HMT in meteorites.

Lastly, we changed the word “volatiles” in the title with “prebiotic chemistry” as suggested.

[Additional comment 1] I believe "Rubin, M., Bekaert, D. V., Broadley, M. W., Drozdovskaya, M. N., & Wampfler, S. F. (2019). Volatile species in comet 67P/Churyumov-Gerasimenko: investigating the link from the ISM to the terrestrial planets. ACS earth and space chemistry, 3(9), 1792-1811." would be a good reference to cite in order to introduce the concept of the ISM-comet-Earth connection

[Reply] We added the following sentence with some appropriate references into Line 51 (note: the cited references are numbered appropriately in the revised manuscript): “... , similar to other interstellar molecules (Ehrenfreund & Charnley 2000; Rubin et al. 2019).”

[Additional comment 2] I would delete "likely" L37

[Reply] Deleted as suggested.

[Additional comment 3] I would change "volatile" for "prebiotic" L38

[Reply] We do not think formaldehyde and ammonia are “prebiotic molecules”. Then we changed “volatile molecules” with “important molecules for prebiotic chemistry”.

[Additional comment 4] I would add "at" before "~10K" L43

[Reply] Added as suggested.

[Additional comment 5] Please add a reference to the sentence L45-47

[Reply] Bernstein et al. (1995) and Muñoz Caro et al. (2003) A&A, 412, 121-132 are cited as Refs. 15 and 19, respectively.

[Additional comment 6] Please reformulate "its detection in the presence of deep N-H, C-H, and C-N bands in ices would complicate its definitive identification". I guess what you want to say is that the presence of deep N-H, C-H, and C-N bands in ices would prevent straightforward identification of HMT

[Reply] We prefer the original expression here.

[Additional comment 7] Maybe remove the "s" from "detections" L56

[Reply] Deleted as suggested.

[Additional comment 8] I would remove "for the extraction processes to search for HMT" L64

[Reply] We deleted "to search for HMT" here.

[Additional comment 9] L73-76: can you show the spectra as well?

[Reply] We added a Supplementary Table 1 which contains a comparison of the retention time and the measured m/z of HMT standard reagent and that in Murchison under the three different analytical conditions.

[Additional comment 10] L81: Why did you not identify these as well for the two other meteorites?

[Reply] For Tagish Lake, due to the smaller sample abundance (0.5 g) than Murchison (2 g), the isotopologues were below the detection limit. For Murray, since the concentration

of HMT itself is much lower than Murchison, its isotopologues might also be low, which was below the detection limit too.

[Additional comment 11] L103-105: can you please consider making a figure to show how reasonable this agreement is? (and add this to the supplements?)

[Reply] We made a new figure as requested by Reviewer 3. In this figure, the relative abundances of HMT-CH₃, HMT-OH, and HMT-CH₂OH derived from Muñoz Caro et al. (2004) are compared with those in Murchison. We added this figure as Supplementary Figure 5 in the revised version.

[Additional comment 12] L106: delete "the" or add "aliquot" after Murchison

[Reply] "the" is deleted as suggested.

[Additional comment 13] L109: change "was" for "is" to be consistent with the previous sentence

[Reply] Changed as indicated.

[Additional comment 14] "lower abundances or higher loss rates" L112 can you precise whether you are talking about HMT, amino acids, both?

[Reply] We are talking about HMT here. Then we added "... of HMT" into Line 115.

[Additional comment 15] L114-115: how the absence of an obvious trend can support a weak correlation in terms of their formation history? I suggest putting "no obvious correlation" or add more details if you want to suggest there is some correlation despite the absence of evidence for it

[Reply] We changed the words "suggesting a weak correlation" with "suggesting no obvious correlation".

[Additional comment 16] Maybe consider changing "processes" by "degradation" L119

[Reply] Changed as suggested.

[Additional comment 17] L120: this sentence does not need a "," was it "expected" or "observed"? please be specific

[Reply] We deleted “,” here. Since we do not have standard reagents of alkyl-imidazoles, we cannot exactly assign these species; however, based on the trend on the mass chromatograms, their presence is strongly “expected” in Murchison and Murray.

[Additional comment 18] L122: I am confused. you say L119 that these molecules are "proposed as the products after 119 the hydrothermal processes of HMT" and then use their abundances to talk about the extent of "thermal alteration" L122. It is important to distinguish hydrothermal (i.e., aqueous) alteration and thermal alteration (which does not necessarily requires aqueous conditions). Please be specific

[Reply] We changed the word “thermal” with “hydrothermal” here. Also, following to the comment by Reviewer 1, we modified this sentence as follows: “**These results do not contradict with the assumption that Tagish Lake, at least the specimen used in the present study, could have experienced hydrothermal alteration less extensive than Murchison and Murray on their parent bodies.**”

[Additional comment 19] L119-122: can you refer to a figure/reference?

[Reply] We referred Supplementary Figures 7-9 (Supplementary Figures 5-7 in the original version) here. As for the degree of hydrothermal alteration for each meteorite, this is our assumption derived from the present result. So no citations here.

[Additional comment 20] L123-124: this is a personal opinion, but I really don't like presenting a hypothesis as a question. I would suggest saying "one possibility is that" or something along these lines

[Reply] We modified the sentence as follows: “... **one possibility is that some of the HMT and its derivatives can form amino acids during routine amino acid extraction and workup.**”

[Additional comment 21] L126-128: there are a couple of studies that have investigated

the potential link between HMT and amino acids, and I am aware that this one is for instance coming out

V. Vinogradoff, L. Remusat, H. McLain, J. Aponte, S. Bernard, G. Danger, J. Dworkin, J. Elsila, M. Jaber (2020) "Impact of phyllosilicates on amino acid formation under asteroidal conditions". ACS Earth and Space Chemistry. In press

Maybe you want to add more reference about this

[Reply] We modified this sentence with the addition of the suggested reference as follows (these references are numbered in the revised manuscript): "In fact, acid hydrolysis of HMT-containing organic mixtures yielded amino acids, and the role of HMT for amino acid formation has been investigated well in recent studies (Oba et al. 2016, 2017; Vinogradoff et al. 2020)."

[Additional comment 22] L129: remove "naturally"

[Reply] Deleted as suggested.

[Additional comment 23] L131: add a coma after meteorites

[Reply] Added as suggested.

[Additional comment 24] L134-135: please merge the two sentences

[Reply] We prefer the present style since we discuss different topics in these two paragraphs.

[Additional comment 25] "analyzed" L380 – please check that you are consistently being using english vs. american

[Reply] The word "analyzed" was changed with "analysed".

[Additional comment 26] L140: asteroidS

[Reply] Changed as suggested.

[Additional comment 27] L141: at 100°C is it water vapor? What was the pressure of

your experiment?

[Reply] The precise pressure could not be determined at relatively low pressure ranges (1-3 bar) in the present setup, but should be between 1 and 2 bar. This means the water is liquid.

[Additional comment 28] L148: "gas phase of the solar system" are you exclusively talking about the protoplanetary disk period of time? Or if this happens today, it would be desorption to vacuum

[Reply] We mean both here. If HMT is located in the surface of asteroids, it may be partially sputtered out by UV photons, cosmic ray, etc. into vacuum. On the other hand, HMT at the subsurface of asteroids may be protected from such desorption processes.

[Additional comment 29] L154: you can also add reference to Vinogradoff's work here

[Reply] I do not think that citing Vinogradoff 's work here since the content in Line 154 has not been mentioned in her previous works. She did not analyze meteorites and did not discuss the delivery of HMT to the Earth. Rather, she discussed the degradation of HMT leading to the formation of other molecules such as amino acids. Instead, we added Vinogradoff et al. (2020) into Line "153" in the original manuscript (Line 164 in the revised one) with the addition of the word "amino acid" as a product of hydrothermal degradation of HMT.

[Additional comment 30] L155: "dust" are you specifically talking about interplanetary dust particles? Please be specific

[Reply] We mean interplanetary dust particles. Modified appropriately.

[Additional comment 31] L163-164: can you please add some information about the past location of the snow line during the evolution of the solar system. The location of the snowline in the protoplanetary disk is a direct function of the stellar accretion rate and is expected to move inward as the accretion rate decreases with time (Bitsch et al., 2015). Based on astronomical observations, stellar accretion rates are estimated to have decreased on average from 10^{-8} Ms/y at 1 My to $1-5 \cdot 10^{-9} \text{ Ms/y}$ at 3 My ($\text{Ms} =$ the solar mass) (Hartmann et al., 1998), inducing the snowline to drift from 3 AU to 1 AU

(Bitsch et al., 2015). Such a snowline drift implies that water ice grains were ubiquitous in the inner Solar System, and formed abundant ice-rich planetesimals.

I'm not asking you to add all of this stuff, but at least add provide the reader with some intuition regarding the spatial evolution of the snow line during the evolution of the PPD

[Reply] Although there are some calculations about the location of “water snow line” depending on the time after the solar system formation, there is no research which definitely determined the location of the “ammonia (formaldehyde) snow line”, which should be different from water snow line. So to avoid any uncertain discussion about the location of “snow line”, we deleted the sentence “This is in particular applicable... asteroid belt is located”. Instead, we added the following sentence “if the temperature of the formed asteroid exceeds the desorption temperature of both molecules.”

[Additional comment 32] L169: what do you mean by workup? lab processing?

[Reply] It's true. We added “laboratory” before “workup”.

[Additional comment 33] L171: I suggest changing "volatiles" for "prebiotic precursors" and add "inner" before "solar system"

[Reply] Modified as suggested.

[Additional comment 34] L179-180: "The sample quality was previously evaluated for amino acids" what does it mean? It just means you analyzed amino acids? How does it relate to the sample quality?

[Reply] “Sample quality” means a degree of contamination here. We added “(i.e. a degree of contamination)” after the word “quality”.

[Additional comment 35] L256: space missing after 1999

[Reply] A space was inserted.

[Additional comment 36] I would remove the molecular structure from Fig. 2 as it looks like it only concerns the first spectrum.

[Reply] Removed as suggested.

REVIEWER COMMENTS

Reviewer #1 (Remarks to the Author):

I appreciate the authors revising their manuscript well. I think the manuscript is almost ready for publication. I have one more thing I should comment on this manuscript.

The authors emphasize on interstellar origin of HMT. However, as mention in line 105 (and only here), HMT was only detected after warming up to room temperature from irradiation experiments of interstellar ice analogues. Vinogradoff et al. A&A 551, A128 (2013) pointed out that the warming-up process plays a key role for HMT formation. Thus, it is not appropriate to call it "interstellar HMT" e.g., as in line 182. The warming up process should be mentioned in e.g., lines 32-35, 44-47 and 50-51, as well.

Minor issues

L152, I prefer "in asteroids" instead of "on asteroids" since hydrothermal processes took place inside asteroids.

The appendix may be moved to supplementary.

Reviewer #2 (Remarks to the Author):

dear Authors,

Please find my review in the attached pdf document.

Sincerely.

Reviewer #3 (Remarks to the Author):

I have now read the response to reviewers document and revised version of the manuscript entitled "Hexamethylenetetramine in meteorites: a precursor of prebiotic chemistry in the inner solar system", by Oba et al.

The authors carefully addressed my comments, and I would like to thank them for doing so .

I am therefore happy to recommend the manuscript for publication in Nature Communications.

David V. Bekaert

Replies to comments by Reviewers

We appreciate the constructive review comments from three Reviewers on our revised manuscript (NCOMMS-20-26492A) entitled “Hexamethylenetetramine in meteorites: a precursor of prebiotic chemistry in the inner solar system”. We carefully read through all of the reviews and modified the manuscript based on their comments accordingly. The changes we made are noted in **red font** in the revised manuscript and supporting information. Our replies to each comment (Times New Roman) are denoted below following to the reviewer’s comments (Arial). Note that in our replies below, we denoted Line numbers in the present version of the manuscript.

Reviewer #1 (Remarks to the Author):

Comment 1. I appreciate the authors revising their manuscript well. I think the manuscript is almost ready for publication. I have one more thing I should comment on this manuscript.

The authors emphasize on interstellar origin of HMT. However, as mention in line 105 (and only here), HMT was only detected after warming up to room temperature from irradiation experiments of interstellar ice analogues. Vinogradoff et al. A&A 551, A128 (2013) pointed out that the warming-up process plays a key role for HMT formation. Thus, it is not appropriate to call it “interstellar HMT” e.g., as in line 182. The warming up process should be mentioned in e.g., lines 32-35, 44-47 and 50-51, as well.

[Reply] Thank you very much for the positive comments on our revised manuscript. We modified the expression “interstellar HMT” with simply “HMT” in the revised manuscript. In addition, we modified the sentence as follows: “Based on laboratory experiments simulating photochemical **and thermal** reactions of interstellar and cometary ice analogues (at ~10 K) ...” (Line 47).

[Minor comment 1] L152, I prefer “in asteroids” instead of “on asteroids” since hydrothermal processes took place inside asteroids.

[Reply] Modified as suggested (Line 166).

[Minor comment 2] The appendix may be moved to supplementary.

[Reply] Appendix moved to supplementary as suggested. The appendix figure was shown as Supplementary Figure 1. Accordingly, the figure numbering was modified.

Reviewer #2 (Remarks to the Author):

[Comment 1]

Dear Authors,

Thank you for your answer about my comments on your paper, I have read all of them and I am almost satisfied. But I still believe that you need to perform mass fractionation. I understand that you may have no more product or no more time, but this is a very important result, (detection of HMT) that some of us were waiting for very long time and we want to be sure that this is not another isomer. The mass fractionation is a sinequanone condition in mass spectrometry analysis and since you can do it with your Orbitrap it should not be avoided. Please look at the paper by Ruf et al, 2019 <https://www.mdpi.com/2075-1729/9/2/35>. As you will see, there is for one exact same mass the coexistence of around 30 different isomers, analyzed here in Murchison extract, so the same meteorite as you. Since you used the exact same analytical method, don't you think you can also have such existence of multiple isomers at the exact same mass as HMT ? When searching for isomers of HMT, with the same mass, on Pub chem, there are around 2000 occurrences (<https://pubchem.ncbi.nlm.nih.gov/#query=C6N4H12&page=1>) meaning 2000 isomers (!). of course a lot of them will have a different retention time by HPLC than HMT, and should not have the good RT + exact mass requested. However, there are structures like these two one as follow, that are very, very similar to HMT (position of N) and might have the exact same retention time (!). they are may be less stable than HMT structure, however, since multiple isomers can exist for the same exact mass (Ruf et al, 2019), we have to be sure that it is really HMT, and only the MS-MS can provide such information.

You are aware that confusion happened for the nucleobases identification (Oba et al, 2019 vs Ruf et al, 2019 <https://doi.org/10.3847/2041-8213/ab59df>), based on the exact same method, so why not using MS-MS since you can? I want to believe that there is some HMT in meteorites, really, but this third proof (fractionation of the ion at 141.1135) has to be performed to confirm it by analytical method. Also I think it is mandatory for a publication in Nature journal. Same fractionation should be done on the HMT-derivatives, searching for mass fragment similar to HMT, as Danger et al, 2013 did (<https://doi.org/10.1016/j.gca.2013.05.015>). If not, and as you agreed, this is speculation to assign the mass to the HMT-derivatives, and you may want moderate such identification in the text to avoid confusion. This paper fully deserves publication, and it is my sincere impression that the quality of this first HMT detection work in meteorites can be improved if applying the major comments recommended above, and other below. If further questions, please feel free to contact me at vassilissa.vinogradoff@univ-amu.fr. Sincerely, Vassilissa

[Reply] We appreciate the comments by Reviewer 2, Dr. Vinogradoff, on the necessity of MS/MS measurements to firmly assign HMT in our samples. We understand that the number of structural isomers increases with increasing the number of atoms in a molecule and MS/MS measurements are often helpful to identify a specific molecule in a mixture of its structural isomers. However, in the present case, only one peak appeared on the mass chromatogram of the meteorite extracts at $m/z = 141.1135$ (within 3 ppm) with a retention time that matched the HMT standard reagent under three different analytical conditions (i.e. InertSustain PFP, Hypercarb, or InertSustain Amide as a separation column). These experimental results provide high confidence that the observed peak is derived from HMT.

Nevertheless, we performed the MS/MS analysis of the Murchison meteorite extract as suggested and compared the results with that of the HMT

standard reagent (please see Results, Methods and Supplementary Figure 3 in the revised manuscript for further details). The observed similarity in the MS/MS mass fragmentation pattern generated from the parent peak of HMT ($M + H^+ = 141.11$) between the standard reagent and the Murchison extract clearly indicates that HMT is present in the Murchison extract. For the Tagish Lake and Murray meteorites, we were unable to perform MS/MS measurements due to the low concentration of HMT in the extracts.

[Appendix: Validation of potential HMT isomers mentioned by Reviewer 2]

Compound (i):

Name: 1,3,5,7-tetraazatricyclo[5.1.1.1^{3,5}] decane

Chemical Formula: C₆H₁₂N₄

Exact Mass: 140.11

We could not locate the suggested molecule in any database of literature references and chemical reagent suppliers. The molecular electrostatic potential (MEP) of the entire molecule and individual C, N and H are different from HMT, suggesting that the molecular polarity of the “1,3,5,7-tetraazatricyclo[5.1.1.1^{3,5}] decane” differs from the HMT behavior on the present chromatographic separation of inertSustain PFP, Hypercarb, InertSustain Amide. Please see Supplementary Figure 1.

Molecular electrostatic potential of “1,3,5,7-tetraazatricyclo[5.1.1.1^{3,5}] decane”

Compound (ii):

as

(2S, 6R)-

(2R, 6S)-

Name:

(2R,6S)-dihydro-1H,3H,5H-2,6-methano[1,2,4]triazolo[1,2-a][1,2,4]triazine
and

(2S,6R)-dihydro-1H,3H,5H-2,6-methano[1,2,4]triazolo[1,2-a][1,2,4]triazine

Chemical Formula: $C_6H_{12}N_4$

Exact Mass: 140.11

The MEPs of these “enantiomer” molecules and individual C, N and H are different from HMT, suggesting that the molecular polarities “both of (2R,6S)- and (2S,6R)-” differ from the HMT behavior on the present chromatography of inertSustain PFP, Hypercarb, and InertSustain Amide. Additionally, there were no enantiomers detected in our analyses, i.e., twin peaks of those stereoisomers. Please see Figure 2.

Therefore, we conclude that these candidate HMT isomers suggested by the reviewer 2 are not possible.

Molecular electrostatic potential of “(2R,6S)-dihydro-1H,3H,5H-2,6-methano[1,2,4]triazolo[1,2-a][1,2,4]triazine”

[Comment 2] Please find below other comments on the manuscript, line by line. I apologize in advance for being so meticulous, but this is a peculiar subject for me too.

-Title : I think the title is not reflecting the work you have done. I think you should consider a title such as : “first detection of HMT in meteorites, clues for organic matter evolution in solar system” or “First detection of HMT in meteorites, : a precursors of prebiotic compounds in the inner solar system bodies”. Since your work is centred about its detection, it should appear on the tittle, more than the fact that it is a precursors of prebiotic compounds, which is not your work.

[Reply] We modified the title as follows: “**Detection of** hexamethylenetetramine in meteorites: a precursor of prebiotic chemistry in the inner solar system.”

[Comment 3] -Line 31, references for organic molecules formation in asteroids can be more appropriate, such as Cody et al, 2011 (your ref 36), Kebukawa et al, 2013

(<https://iopscience.iop.org/article/10.1088/0004-637X/771/1/19/meta>) ,
Vinogradoff et al, 2018 (your ref 23) and Vinogradoff et al., 2020
(<https://doi.org/10.1016/j.gca.2019.10.029>) .

[Reply] We do not think the replacement of the references in the introductory

part is necessary. Instead, we added the suggested references and the other related papers in the Supplementary Information together with the Supplementary text.

[Comment 4] - line 45: I suggest to adapt the sentence as follow to be more exact “*Based on laboratory experiments simulating photochemical and thermal reactions of interstellar and cometary ice analogues (at ~10 K) **initially made of** observed molecules such as water (H₂O), ammonia (NH₃), and methanol (CH₃OH), HMT ...*”

[Reply] Modified as suggested (Lines 47-48).

[Comment 5] -line 47, (up to 60% in the total organic products) □ (up to 60% **weight**) in the total organic products.

[Reply] We modified as “(up to 60% by weight) in the total organic products.” (Line 50)

[Comment 6] -line 48 “photochemical products” refer to product formed after irradiation, at 10K, while HMT is formed at much higher temperature and is the result of thermal processes.

(<https://pubs.rsc.org/en/content/articlelanding/2012/cp/c2cp41963g/unauth#!divAbstract>) I suggest to remove the term “photochemical” in this sentence.

[Reply] Deleted as suggested.

[Comment 7] -line 49: add ref 17, which explain why methanol is necessary.

[Reply] Added as suggested (Line 52).

[Comment 8] -line 51 similar  similarly

[Reply] Grammar correct, change not needed.

[Comment 9] -line 52, HMT has been search in molecular clouds and diffuse ISM
<https://academic.oup.com/mnras/article/298/1/131/976217>

You may want to cite this work.

[Reply] Thank you very much for introducing the reference paper. However, as have been already replied to the comment by Reviewer 3 in the previous version of our replies, we prefer not to reference previous studies which did not detect HMT in meteorites or interstellar clouds.

[Comment 10] -line 55 : the main problem is the presence of silicate, in the exact same spectral range (10 μm) as the major bands of HMT, more than the ice itself.

[Reply] We added the following sentence into the text: "..., **as well as the presence of a strong silicate band at 10 μm ,**" (Lines 58-59).

[Comment 11] -line 57 : HMT has been proposed but it is likely not the major source. The following reference is more complete that Cottin et al.

<https://www.sciencedirect.com/science/article/abs/pii/S0032063305001406>

[Reply] We rephrased the sentence as follows: "However, HMT has been postulated to be **one of the extended sources** of NH_3 and HCN in comets." , (Line 61). The reviewer 2 may understand that we are focusing the meteoritic HMT profiles in the present report.

[Comment 12] -line 61 " HMT is fragile" please add "in liquid water" Because as a solid in cometary-analogue, it can remain stable up to 450-500 K. Please cite Your ref 34 and

<https://www.sciencedirect.com/science/article/abs/pii/S0019103513002388>

[Reply] We deleted "a fragile molecule, particularly" to describe the property of HMT more precisely.

[Comment 13] -line 62 for acidic conditions you can cite the pioneer work of : Hulett, H. R.; Wolman, Y.; Miller, S. L.; Ibanez, J.; Oro, J.; Fox, S. W.; Windsor, C. R. Formaldehyde and Ammonia as Precursors to Prebiotic Amino Acids. Science 1971, 174, 1038–1041.

Replies: NCOMMS-20-26492A

[Reply] The suggested paper is in response to the previous publication by Fox and Windsor (1970) published in *Science*. In fact, the acid hydrolysis of HMT was performed by Wolman, Miller, Ibanez, and Oró (*Science*, vol. 174, page 1039), not by others. Therefore, we consider that citing this discussion as a pioneering work of the acid hydrolysis of HMT may not be appropriate in the context.

[Comment 14] -line 70 without MSMS I would moderate all occurrences of “ion mass 141.1136 is HMT” by “which likely correspond to”

[Reply] This change is not necessary now since we now include MS/MS confirmation of the HMT peak in the Murchison extract. Please see the description of MS/MS results in the main text.

[Comment 15] -line 73 when you compare to the HMT standard reagent, have you measured it after the same exact procedure as described in line 215-220 in the methods ? This could also allow you to have an analytical yield for HMT analysis with your method. In fact, it is not clear in the manuscript if you have done that or not.

[Reply] We have measured the HMT standard reagent through the same procedure as the meteorites and confirmed that the recovery was more than 90%. This finding was added to the Method section.

[Comment 16] -Line 83 “loss of HMT is negligible” please explain it a bit more in the supplementary material or the method, also you should add it in the method section if you have done what I suggested in my previous comment (the line 73)

[Reply] Based on experiments as discussed in the previous response, the loss of HMT during the extraction procedure was less than 10%.

[Comment 17] -line 85 please moderate “well consistent with **possible** HMT-derivatives”

[Reply] No need to rephrase here since we did not mention that we detected the HMT derivatives and only state that the m/z values of the observed peaks

were consistent with that of HMT derivatives.

[Comment 18] -line 95 you can add at the end of the sentence “ *The absence of these species on the mass chromatograms for the HMT standard reagent (Supplementary Fig. 4) indicates that these are likely not formed during workup or clusters or N-functionalizations formed by ESI and so should be indigenous to the meteorites samples.*

[Reply] Modified as suggested (Lines 105-106).

[Comment 19] -line 96 I suggest to add “without authentic standards and **if their identification are confirmed**” (with MSMS ?)

[Reply] We modified the sentence as follows: “Without authentic standards, an estimate of their **possible** abundances assumed the same...” (Lines 106-107).

[Comment 20] -line 96 sentence are not correct; I suggest: “[..] **we estimate their abundance assuming the same ionization** [..]”

[Reply] No need to modify the sentence here.

[Comment 21] -line 100 : if no MSMS I would moderate this sentence, arguing that yes, the ion mass detected is indigenous to the meteorites, and could correspond to HMT.

[Reply] We showed that the MS/MS results strongly supported the evidence of indigenous HMT; therefore, there is no need to rephrase this sentence due to the same reason described in our reply to the Comment 1.

[Comment 22] -line 104 Again, this is not sure since no standard of these HMT derivatives exist. So I would prefer have been “**tentatively identified**”

[Reply] We have already mentioned in Lines 102-103 that “these assignments are the most likely but other isomers cannot be excluded”. So, we do not need to add the words suggested by Reviewer 2.

[Comment 23] -line 110, you compare HMT abundance to sugars and amino acids, but the sample was not the same? even not the analytical method. May be it should be mentioned.

[Reply] Since the appropriate references are cited here, we do not need to mention the details of the analytical methods.

[Comment 24] -line 110 : the Tagish lake sample, do you know from which mineralogy it came from ? highly altered ? or not ?

[Reply] Unfortunately, we do not have the petrologic type information for the Tagish Lake meteorite sample used in this study.

[Comment 25] -line 102 -> is **also** in the range

[Reply] We added the word as suggested (Line 125).

[Comment 26] -line 115 : “*Unrelated to glycine*”, but why not with alkyimidazole ? . Add a precision here.

[Reply] We modified the sentence in Lines 128-129 as follows: “...lower abundances or higher loss rates of HMT, **which may partly be related** to the formation of soluble organics. **For example**, Supplementary Fig. 6 shows ...”.

[Comment 27] -line 117 [*there seems no obvious ...*] Even if I understand the point, I think you need to add references explaining why you are comparing HMT abundance with glycine. There is no study in my knowledge linking directly these two compounds. And my experiments indeed show that it is impossible to directly link both compounds because of hydrothermal alteration conditions and minerals and many other chemical reactions that can also produce glycine rather than HMT . Ref 27.

[Reply] The reason why we selected the amino acid glycine for comparison to the HMT abundance is glycine is one of the most common and abundant amino acids found in carbonaceous meteorites. We do not imply a direct link

between both molecules, but since the formation of glycine from HMT is possible (Wolman et al. 1971, Science; Vinogradoff et al. 2020, ACS Earth Space Chem.), a comparison between the abundances of glycine and HMT in meteorites given in the paper is justified.

[Comment 28] -line 124-125 [these results do not contradict..] I would moderate this statement by the fact that there is no proof that Tagish lake, Murray, and Murchison accreted the exact same concentration of HMT (if so) in their parent bodies. It is somehow written in line 133. But I think you should more insist on this point to explain the difference you observed between the HMT abundance. Another idea would be to say that in Murray and Murchison it is possible that HMT already partially decomposed into alkylimidazole, as shown by me (23 and <https://doi.org/10.1016/j.gca.2019.10.029>) while the HMT in the Tagish lake sample has been somehow preserved. but further investigations on the mineralogy of the sample you get from Tagish lake are needed to conclude on the potential low alteration.

[Reply] This point was already been pointed out by Reviewer 1 in the first review, and we modified as it is. Without the petrological data, we are not sure whether the Tagish Lake specimen used in the present study has experienced less extensive hydrothermal alteration than other two meteorites. We just mean that the observed trend does not contradict such an assumption. As has been already mentioned in Lines 157-159, there are still a number of uncertainties on the different HMT concentrations between meteorites. We just give a possible hypothesis to explain the data.

[Comment 29] -line 127 to 133 : This is not correlated with your results, this is to argue for amino acids analysis, not HMT since you haven't used the same extraction method. I think this part, from line 127 to 145 has to be rewritten. The fact that HMT can produce amino acids during routine extraction is out of the scope. I guess you want to explain the loss of HMT during hydrothermal alteration ?. So in this paragraph, I suggest to have something related to the fact that HMT and derivatives can also be transformed during hydrothermal alteration into amino acids, as shown by 27.

[Reply] We believe Reviewer 2 may not understand what we mean here. In

most or all of previous meteoritic amino acid studies, very harsh extraction conditions including high temperature and acid have been applied. Therefore, we suspect that under these conditions any meteoritic HMT would have been hydrolyzed and may have resulted in the formation of amino acids and other products. However, we expect that this would not be the case under the extraction conditions used in this study as explained in Line 144-147.

[Comment 30] -Line 131 I am sorry but you can't compare amino acid abundance in meteorites, measured by other techniques, with HMT abundance. There are plenty other precursors that can explain amino acid formation in meteorites, the link with HMT is too far. The difference in HMT abundance can be explained by other factors.

[Reply] Again, we do not intend to link both molecules directly, but just compare the representative amino acid in meteorites with the present target, HMT, both of which may have a link, even if it isn't directly, in the same meteorite. We believe that Reviewer 2 may be confusing laboratory simulation experiments with analyses of meteoritic organics.

[Comment 31] -line 140 : I suggest instead of "*Although it is not clear whether such precursors can contribute to the formation of HMT in aqueous solutions without acid and high-temperature treatment at room temperature*"

"Considering the harsh conditions to have formaldehyde and ammonia from meteorite organic compounds (acid and high-temperature treatment) we expect that [..]"

[Reply] We believe that Reviewer 2 may not understand what we mean here. We discuss the possibility that HMT is formed from free ammonia and formaldehyde, which may be present in meteorites.

[Comment 32] -Line 144-145 : Well, definitively I think this part has to be rewritten, since at the end it gives no clue to answer the abundance differences. I think it is related to : HMT abundance accreted, hydrothermal alteration conditions (time, temperature) and minerals. Please see my article with the minerals:

<https://doi.org/10.1016/j.gca.2019.10.029>

[Reply] Even if those parameters are necessary to constrain the HMT abundance in meteorites, it is not easy to reproduce the observed HMT abundance. In addition, other parameters such as photolysis/radiolysis of HMT on meteorite parent bodies cannot be ignored. There are several factors and unknowns about the origin of HMT in meteorites that are out of scope for this paper and should be the focus of follow-on studies.

[Comment 33] -line 154-155 Very nice results about the deuterated HMT ! and I think very important and should be more in the front or put in another paper (? - see my comment below).

[Reply] We have a plan to publish these results in a separate paper.

[Comment 34] -line 165 please add the ref with the phyllosilicates

<https://doi.org/10.1016/j.gca.2019.10.029>

[Comment 35] -line 172 add ref 23, 27 and

<https://doi.org/10.1016/j.gca.2019.10.029> and Kebuwaka et al. 2013

[Reply to both above] Added as suggested.

[Comment 36] -174 : H₂CO is also primarily formed by the photodegradation of methanol. See Oberg 2009

<https://www.aanda.org/articles/aa/abs/2009/36/aa12559-09/aa12559-09.html>

[Reply] We do mention the formation of H₂CO in interstellar ices where the incident UV flux is not high. The main route to the formation of H₂CO in these environments is well-known to be the hydrogenation of CO through quantum tunneling. Also, CH₃OH is produced from the hydrogenation of H₂CO. See Hama & Watanabe (2013) Chem. Rev. for further details.

[Comment 37] -line 176 [*if the temperature of the formed asteroids*] this part of the sentence is not at the right place, since the process you described happens before asteroids accretion. Please remove or rephrase.

[Reply] This statement is not necessarily true. There are some astronomical objects in the outer region of our solar system where the temperature is ~50 K

or below, e.g. TNOs. In addition, as pointed out by other two Reviewers, a potential snowline drift may cause variations in the temperature of such objects, which may lead to the desorption of both molecules depending on the temperature. We do not limit our discussion only to the asteroid belt at ~2-3 AU from the Sun, and would like to be inclusive of environments across the entire solar system.

[Comment 38] -line 178 : In fact, we have shown that formaldehyde and ammonia are precursors for HMT, so in a way they are transformed and preserved into the form of HMT (that is stable until high temperature as a solid, 400 K), and then formaldehyde and ammonia are released during hydrothermal alteration from HMT on the parent bodies. Ref 17 and <https://pubs.rsc.org/en/content/articlelanding/2012/cp/c2cp41963g/unauth#!divAbstract> and 23.

[Reply] We added ref 23 here (Line 192).

[Comment 39] -line 181 : [*from both molecules, although they have been identified in carbonaceous meteorites upon hydrothermal treatment at around 100 °C or acid hydrolysis*] -> Already said before. I suggest to remove this sentence from here. Instead, the idea is that HMT is in equilibrium with ammonia and formaldehyde, its degradation can be slower because of this equilibrium (destruction <-> formation).

[Reply] We repeated this sentence here intentionally since this is a very important suggestion from the present study and this was completely ignored in previous studies. We then modified the sentence as follows: "... from both molecules **if they are really present, which could keep the HMT concentration relatively constant. However,** they have..." (Lines 194-195).

[Comment 40] -Line 184 I think in this sentence, a conclusion is missing, I propose something such as [*As such, it will be challenging to constrain the location of HMT formation* **but its presence in interstellar ice analogues (ref 15.16.17) can be a good start to explain its presence in meteorites.**]

[Reply] We modified the sentence as follows: "As such it will be challenging

to constrain the location of HMT formation **but its presence in the processed interstellar ice analogues¹⁵⁻¹⁸ can be a good indicator to explain its presence in meteorites. Hence,** the presence of HMT...” (Lines 198-200).

[Comment 41] -line 233 please precise when you said HMT if it is the standard or the meteorite samples. I suggest to use the term “meteorite extract” instead of HMT in the method if this is not the standard.

[Reply] We changed the word “HMT” with “The Murchison extract” as suggested (Line 251). The HMT standard reagent was also analyzed by the same exact same methods.

[Comment 42] -line 265 : Really very interesting experiments, have you planned to do more like that and to publish it in a separated paper ? in fact I am not sure this is request/needed here for the manuscript, and I would suggest to keep it for another publication. What do you think ?

[Reply] As has been replied to Comment 33, we have a plan to summarize the related result and publish it in a separate journal. We just showed a tiny part of the experimental results in the present manuscript, so we think it does not matter. We hope the related, more systematic results may be reported in (near) future.

[Comment 43] Table 1 : summary of “possible HMT-derivatives”

[Reply] Modified as suggested.

[Comment 44] Figure 2 could you indicate the column used, and precise the RT for the peak detected.

[Reply] An InertSustain PFP column was used. This information was added to the caption. The retention time was added to the Figure.

[Comment 45] Figure 3, same here as figure 2.

[Reply] An InertSustain PFP column was used. This information was added

to the caption. The retention time of possible derivatives were not added to the Figure since we cannot assign which peaks are derived from the HMT-derivatives.

[Comment 46] Supl figure 3 : We need more explication on that experiment, please add a supplementary text.

[Reply] We added the following sentence into the caption of Supplementary Fig. 5: “The sample was analysed first without purification (concentration: X-axis). After that, the same sample was dried and processed in the same manner with the addition of the purification procedure (concentration: Y-axis).”

[Comment 47] Supl table 1, what about the other meteorite samples ? the ion mass peak in tagish lake seems to be shift.

[Reply] We did not analyze the Murray and Tagish Lake extracts using Hypercarb and Amide columns since the cross-check has been done for the Murchison extract. The possible reason for the peak shift observed in the Tagish Lake extract has been explained in the caption of Figure 2.

Reviewer #3 (Remarks to the Author):

I have now read the response to reviewers document and revised version of the manuscript entitled "Hexamethylenetetramine in meteorites: a precursor of prebiotic chemistry in the inner solar system", by Oba et al.

The authors carefully addressed my comments, and I would like to thank them for doing so.

I am therefore happy to recommend the manuscript for publication in Nature Communications.

David V. Bekaert

Replies: NCOMMS-20-26492A

[Reply] We appreciate the comments by Reviewer 3, Dr. Bekaert, for his fruitful comments on our manuscript, which improved significantly the revised version.

REVIEWERS' COMMENTS

Reviewer #2 (Remarks to the Author):

Dear Authors,

I appreciate the author revisions, and I think the paper is almost ready for publication in Nature Comm. I have few more suggestions or answer to your replies that you will find in attached. please consider this few more corrections to again improve your paper. Overall this is a wonderful work.

Sincerely

Vassilissa Vinogradoff

Dear authors,

Thank you very much for your extensive reply to my preview comments. I am very glad about the new result from mass fractionation! This is very very great and now we can be sure (also the reader) that HMT is effectively present in Murchison SOM, at least. I strongly suggest you to put this figure 3 currently in Supplementary, in the main text, since it is essential for the paper. In return, you may want to put current figure 3 in the main text in supplementary since the attribution of the isomers remains speculative.

Please find below some answer to your comments on my previous review, and complementary suggestions (each time I am cited you, your text is in *Italic*). Please take my comments as gentle suggestions to improve your paper at the level request by Nature, and also because it is important for our community.

Many time you thought that I don't understand the meaning of your sentences, which is in fact not true, it was in contrary either suggestions or rephrasing, or because words a missing to fully understand what you want to say. Some argument can be more accurate. I think that you can understand that if I do a comment, it is because it is not clear for me. And if it is not clear for me, while I know as much as you the subject, it will be not clear for the reader as well. So this is not criticism, but improvement, and I took some time on your paper for that.

Also I am not an English native speaker, but I think there are some grammar mistakes (I guess editorial office will care of that)

Comments:

*In the caption of new suppl figure 3 you said: *"The retention time of such other species was not exactly the same with that of HMT and its fragments, which supports our conclusion that the peaks with an asterisk are not derived from HMT"*

I don't understand this statement, how can you know that the other species don't have the same retention time since you have fragmented the ion 141.11 ? so it only means that HMT is not the only species (but the major one) at this mass and this is ok. I think this statement should be rewrite or remove. You know that we can have multiple isomers for the same exact mass (Ruf et al, 2019) so it is not surprising that at mass 141.11 you have also other species. This is in contrary reassuring, regarding the large diversity of compounds found in Murchison SOM (Schmitt-kopplin, PNAS, 2010).

*You said that *"For the Tagish Lake and Murray meteorites, we were unable to perform MS/MS measurements due to the low concentration of HMT in the extracts."*

I think you mean "due to small quantity of samples" ? since according to your quantification there is 671 ppb of HMT in Tagish lake... also this should be mention in the method section or in the text when describing MS-MS measurement.

*It was not necessary to search for the isomers I put in my review (thanks !), it was just examples. I think we misunderstood each other and I am sorry for that.

*line 43, you may want to check your ref and add few more that you have cited after and that deserved to be also in that list, since in any case it is not exhaustive. For example, ref 9, doesn't refer to organic molecules (only water), and ref 23 (at least, but also 35 and 27) and ref 39 should be with 13,14. I am sorry but It would be fair to cited all the refs you used later, or not at all (or say :for example see ref...). You can instead cite 32 and 20 that are review papers (but not up to date, obviously).

*Line 43 I suggest to add precision: "it still remains under debate when, where, and how such extraterrestrial molecules in meteorites were abiotically formed" (sentence also in the abstract).

*[Comment 9] -line 52, HMT has been search in molecular clouds and diffuse ISM

<https://academic.oup.com/mnras/article/298/1/131/976217>

You may want to cite this work.

[Reply] Thank you very much for introducing the reference paper. However, as have been already replied to the comment by Reviewer 3 in the previous version of our replies, we prefer not to reference previous studies which did not detect HMT in meteorites or interstellar clouds.

I think it is in contrary useful to show this study where HMT has been tentatively search for in interstellar environment, and not found, it is also a justification of your sentences, and I suggest to add this reference line 60.

*Line 64, ref 23 can may be move just after water because I never used acid hydrolysis method in my samples.

*Line 107, "Without authentic standards, an estimate of their *possible* abundances assumed the same ionization efficiency as HMT"

I still think something is wrong with the grammar in the sentence, assumed → assuming ? and a verb is missing ? I guess the editorial office will correct that.

*Line 122, individual or total amino acids ? I guess "total" has to be remove.

*Line 125 pleas add : "in the range of individual amino acid concentrations"

*And what about amino acids abundances in Murray meteorite? Can you say a word ?

*Line 128 “parent body conditions” please give example in parentheses. I can be obvious for you, but not for the reader. (temperature ? minerals? Degree of hydrothermal alteration ? others ?)

**[Comment 29] -line 127 to 133 : This is not correlated with your results, this is to argue for amino acids analysis, not HMT since you haven't used the same extraction method. I think this part, from line 127 to 145 has to be rewritten. The fact that HMT can produced amino acids during routine extraction is out of the scope. I guess you want to explain the loss of HMT during hydrothermal alteration ?. So in this paragraph, I suggest to have something related to the fact that HMT and derivatives can also be transformed during hydrothermal alteration into amino acids, as shown by 27.*

*[Reply] We believe Reviewer 2 may not understand what we mean here. In most or all of previous meteoritic amino acid studies, very harsh extraction conditions including high temperature and acid have been applied. **Therefore, we suspect that under these conditions any meteoritic HMT would have been hydrolyzed and may have resulted in the formation of amino acids and other products. However, we expect that this would not be the case under the extraction conditions used in this study** as explained in Line 144-147.*

I fully understand what you meant, but it is not clear in the paragraph if this argument is to explain why HMT has never been observed before in meteorites? Or if it is to explain your abundance difference in the results ? or both.. something wrong.

Line 145-146 is very confusing. In fact the sentence in your answer (in yellow now) is clearer than the sentence : “However, that HMT is the origin of amino acids during workup is weakened by Murray, which has a similar abundance of amino acids to Murchison⁸, yet the HMT concentration was lower by about an order of magnitude than Murchison.” (also how Murray can weakened the workup... by →for)

*line 150 : on the other hand ? where is the “on one hand”.. ? please correct.

[Comment 31] -line 140 : I suggest instead of “Although it is not clear whether such precursors can contribute to the formation of HMT in aqueous solutions without acid and high-temperature treatment at room temperature” “Considering the harsh conditions to have formaldehyde and ammonia from meteorite organic compounds (acid and high-temperature treatment) we expect that [..]***

[Reply] We believe that Reviewer 2 may not understand what we mean here. We discuss the possibility that HMT is formed from free ammonia and formaldehyde, which may be present in meteorites.

Again, I completely understood the same things, I just suggest a rephrasing of the sentence which is heavy or lack of some words, for example if you wanted to keep your sentence: “Although it is not clear whether such precursors can contribute to the formation of HMT in our samples after aqueous solutions without acid and high-temperature treatment at room temperature”

Sometimes my comment is because your sentence is not clear enough, or precise enough, but be reassured that in all cases, I understand the meaning. If not I would say it.

*Line 157 related to comment 32. I agree with the authors that many parameters are necessary to constrain HMT abundance and origins, but I think you can give one more example in line 157 that is important, it is the heterogeneity in the organic compounds accreted by the parent bodies of meteorites. Which is another argument that the “heterogeneity in samples from same meteorite” discussed line 148. Compared to this argument (different OM accreted), “parameters such as photolysis/radiolysis of HMT on meteorite parent bodies” is negligible.

You may know that Tagish lake has 99 % IOM (overall) so the abundance of SOM is completely different than in Murchison and Murray. Even if this doesn't explain at all your results, it argues for big differences in the OM accreted.

*Line 186: please check your ref again, please add 23, and 27.

*line 189: ‘both molecules are likely to be lost from grains during warming up phases’. This is partially wrong; it is only if you considered that they will not react with other molecules. So I would suggest to add “if they do not chemically react”. As you know, many chemical reactions happened during the warming up phase, the older statement that it was only for desorption is obsolete.

(<https://www.sciencedirect.com/science/article/abs/pii/S0273117713004183>)

*[Comment 37] -line 176 [if the temperature of the formed asteroids] this part of the sentence is not a the right place, since the process you described happen before asteroids accretion. Please remove or rephrase.

[Reply] This statement is not necessarily true. There are some astronomical objects in the outer region of our solar system where the temperature is ~50 K or below, e.g. TNOs. In addition, as pointed out by other two Reviewers, a potential snowline drift may cause variations in the temperature of such objects, which may lead to the desorption of both molecules depending on the temperature. We do not limit our discussion only to the asteroid belt at ~2-3 AU from the Sun, and would like to be inclusive of environments across the entire solar system.

What you are replying is not related to my comment. I am not discussing the temperature. Your arguments are entirely true, but it was not my comment.;

Please carefully read again your sentence, line 190. You have written:

“both molecules are likely to be lost from grains during warming up phases toward star formation if the temperature of the formed asteroid exceeds the desorption temperature of both molecules”

You are describing warming up phases during star formation, and then you jump to already accreted asteroids. They are two different episodes, no ? first grains are formed in molecular clouds, mixed, irradiated, warmed up (and for sur to much warmed in the inner solar region) before incorporation into asteroids or comets. Your sentence imply that your grains were already accreted into asteroids. I hope to be clearer this time for this comment. Can you rephrase to avoid the confusion please.

*Line 195 “however, they have been “, replace they by ammonia and formaldehyde, when reading we don’t know anymore that it refers to ammonia and formaldehyde.

*line 199: I think you forget one ref, the 19, so ref should be 15-19.

Thank you for the rest of your reply.

V. Vinogradoff

Replies to comments by Reviewers

We appreciate the constructive review comments from Reviewer #2, Dr. Vinogradoff, on our revised manuscript (NCOMMS-20-26492B) entitled “Detection of hexamethylenetetramine in meteorites: a precursor of prebiotic chemistry in the inner solar system”. We carefully read through all of the reviews and modified the manuscript based on her comments accordingly. The changes we made are noted in **red font** in the revised manuscript and supporting information. Our replies to each comment (Times New Roman) are denoted below following to the reviewer’s comments (Arial). Note that in our replies below, we denoted Line numbers in the present version of the manuscript.

Reviewer #2 (Remarks to the Author):

[Comment 1]

Dear authors,

Thank you very much for your extensive reply to my preview comments. I am very glad about the new result from mass fractionation! This is very very great and now we can be sure (also the reader) that HMT is effectively present in Murchison SOM, at least. I strongly suggest you to put this figure 3 currently in Supplementary, in the main text, since it is essential for the paper. In return, you may want to put current figure 3 in the main text in supplementary since the attribution of the isomers remains speculative.

Please find below some answer to your comments on my previous review, and complementary suggestions (each time I am cited you, your text is in *Italic*).

Please take my comments as gentle suggestions to improve your paper at the level request by Nature, and also because it is important for our community.

Many time you thought that I don’t understand the meaning of your sentences, which is in fact not true, it was in contrary either suggestions or rephrasing, or because words a missing to fully understand what you want to say. Some argument can be more accurate. I think that you can understand that if I do a comment, it is because it is not clear for me. And if it is not clear for me, while I know as much as you the subject, it will be not clear for the reader as well. So this is not criticism, but improvement, and I took some time on your paper for that.

Also I am not an English native speaker, but I think there are some grammar mistakes (I guess editorial office will care of that)

[Reply] We appreciate the comments by Reviewer #2, Dr. Vinogradoff, on our revised version. MS/MS spectra were moved to the main text as Figure 3. We prefer the original Figure 3 remains in the main text, as Figure 4.

[Comment 2] *In the caption of new suppl figure 3 you said: “*The retention time of such other species was not exactly the same with that of HMT and its fragments, which supports our conclusion that the peaks with an asterisk are not derived from HMT*”

I don't understand this statement, how can you know that the other species don't have the same retention time since you have fragmented the ion 141.11 ? so it only means that HMT is not the only species (but the major one) at this mass and this is ok. I think this statement should be rewrite or remove. You know that we can have multiple isomers for the same exact mass (Ruf et al, 2019) so it is not surprising that at mass 141.11 you have also other species. This is in contrary reassuring, regarding the large diversity of compounds found in Murchison SOM (Schmitt-koplin, PNAS, 2010).

[Reply] We deleted the sentence as suggested.

[Comment 3] *You said that “*For the Tagish Lake and Murray meteorites, we were unable to perform MS/MS measurements due to the low concentration of HMT in the extracts.*”

I think you mean “due to small quantity of samples” ? since according to your quantification there is 671 ppb of HMT in Tagish lake... also this should be mention in the method section or in the text when describing MS-MS measurement.

[Reply] As you may realize, 671 ppb is the concentration of HMT in the “meteorite”, but this is not equal to that in the Tagish Lake “extract”. We added the sentence into the method section as suggested.

[Comment 4] *It was not necessary to search for the isomers I put in my review

(thanks !), it was just examples. I think we misunderstood each other and I am sorry for that.

[Reply] We expect that there could be some structural isomers which appear at the same retention time under a single analytical condition; however, we still believe that it is unlikely that the retention times of two different structural isomers are exactly the same under the three different analytical conditions.

[Comment 5] *line 43, you may want to check your ref and add few more that you have cited after and that deserved to be also in that list, since in any case it is not exhaustive. For example, ref 9, doesn't refer to organic molecules (only water), and ref 23 (at least, but also 35 and 27) and ref 39 should be with 13,14. I am sorry but It would be fair to cited all the refs you used later, or not at all (or say :for example see ref...). You can instead cite 32 and 20 that are review papers (but not up to date, obviously).

[Reply] Vinogradoff et al. (2018) was added as suggested.

[Comment 6] *Line 43 I suggest to add precision: "it still remains under debate when, where, and how such extraterrestrial molecules **in meteorites** were abiotically formed" (sentence also in the abstract).

[Reply] We do not agree with the Reviewer's suggestion. Not only molecules in meteorites, but also those in other extraterrestrial environments are still targets for a number of researches which are trying to solve their synthetic processes.

[Comment 7] **[Comment 9] -line 52, HMT has been search in molecular clouds and diffuse ISM*

<https://academic.oup.com/mnras/article/298/1/131/976217>

You may want to cite this work.

[Reply] Thank you very much for introducing the reference paper. However, as have been already replied to the comment by Reviewer 3 in the previous version of our replies, we prefer not to reference previous studies which did not detect HMT in meteorites or interstellar clouds.

I think it is in contrary useful to show this study where HMT has been tentatively search for in interstellar environment, and not found, it is also a justification of your sentences, and I suggest to add this reference line 60.

[Reply] Cited as suggested.

[Comment 8] *Line 64, ref 23 can may be move just after water because I never used acid hydrolysis method in my samples.

[Reply] Moved as suggested.

[Comment 9] *Line 107, "*Without authentic standards, an estimate of their possible abundances assumed the same ionization efficiency as HMT*" I still think something is wrong with the grammar in the sentence, assumed assuming ? and a verb is missing ? I guess the editorial office will correct that.

[Reply] We think this is grammatically not wrong. However, if necessary, the editorial office will correct it.

[Comment 10] *Line 122, individual or total amino acids ? I guess "total" has to be remove.

[Reply] The word "total" was removed.

[Comment 11] *Line 125 pleas add : "in the range of **individual** amino acid concentrations"

[Reply] We originally showed the total amino acid concentration for Tagish Lake. However, similar to Murchison, we showed the concentrations of individual amino acids in Tagish Lake. Then, the word "individual" is added here. Accordingly, the amino acid concentration was modified appropriately.

[Comment 12] *And what about amino acids abundances in Murray meteorite? Can you say a word ?

[Reply] We added the following sentence into the text: **While in Murray, the**

concentration of HMT (29 ± 9 ppb) is lower than individual amino acid concentrations (51-2834 ppb) in the same meteorite⁸.

[Comment 13] *Line 128 “parent body conditions” please give example in parentheses. I can be obvious for you, but not for the reader. (temperature ? minerals? Degree of hydrothermal alteration ? others ?)

[Reply] We added “(e.g. temperature, water/rock ratio, etc.)” here.

[Comment 14] *[Comment 29] -line 127 to 133 : This is not correlated with your results, this is to argue for amino acids analysis, not HMT since you haven't used the same extraction method. I think this part, from line 127 to 145 has to be rewritten. The fact that HMT can produced amino acids during routine extraction is out of the scope. I guess you want to explain the loss of HMT during hydrothermal alteration ? So in this paragraph, I suggest to have something related to the fact that HMT and derivatives can also be transformed during hydrothermal alteration into amino acids, as shown by 27.

[Reply] We believe Reviewer 2 may not understand what we mean here. In most or all of previous meteoritic amino acid studies, very harsh extraction conditions including high temperature and acid have been applied. **Therefore, we suspect that under these conditions any meteoritic HMT would have been hydrolyzed and may have resulted in the formation of amino acids and other products. However, we expect that this would not be the case under the extraction conditions used in this study** as explained in Line 144-147.

I fully understand what you meant, but it is not clear in the paragraph if this argument is to explain why HMT has never been observed before in meteorites? Or if it is to explain your abundance difference in the results ? or both.. something wrong.

Line 145-146 is very confusing. In fact the sentence in your answer (in yellow now) is clearer than the sentence : “However, that HMT is the origin of amino acids during workup is weakened by Murray, which has a similar abundance of amino acids to Murchison⁸, yet the HMT concentration was lower by about an order of magnitude than Murchison.” (also how Murray can weakened the workup... by → for)

[Reply] We replaced the following sentence “However, that HMT is the origin of amino acids during workup is weakened by Murray, which has a similar abundance of amino acids to Murchison⁸, yet the HMT concentration was lower by about an order of magnitude than Murchison” with “**Therefore, we suspect that under these conditions any meteoritic HMT would have been hydrolyzed and may have resulted in the formation of amino acids and other products. However, we expect that this would not be the case under the extraction conditions used in this study**”.

[Comment 15] *line 150 : on the other hand ? where is the “on one hand”.. ? please correct.

[Reply] In the previous sentences, we discussed the possible “degradation” of HMT during workup. This is it. After the words “On the other hand”, we discussed the possible “formation” of HMT.

[Comment 16] **[Comment 31] -line 140 : I suggest instead of “Although it is not clear whether such precursors can contribute to the formation of HMT in aqueous solutions without acid and high-temperature treatment at room temperature” “Considering the harsh conditions to have formaldehyde and ammonia from meteorite organic compounds (acid and high-temperature treatment) we expect that [..]*

[Reply] We believe that Reviewer 2 may not understand what we mean here. We discuss the possibility that HMT is formed from free ammonia and formaldehyde, which may be present in meteorites.

Again, I completely understood the same things, I just suggest a rephrasing of the sentence which is heavy or lack of some words, for example if you wanted to keep your sentence: “*Although it is not clear whether such precursors can contribute to the formation of HMT in our samples after aqueous solutions without acid and high-temperature treatment at room temperature*”

Sometimes my comment is because your sentence is not clear enough, or precise enough, but be reassured that in all cases, I understand the meaning. If not I would say it.

[Reply] We understood that Reviewer 2 understood what we meant. Thank you for your suggestion. However, we prefer the original sentence here.

[Comment 17] *Line 157 related to comment 32. I agree with the authors that many parameters are necessary to constrain HMT abundance and origins, but I think you can give one more example in line 157 that is important, it is the heterogeneity in the organic compounds accreted by the parent bodies of meteorites. Which is another argument that the “heterogeneity in samples from same meteorite” discussed line 148. Compared to this argument (different OM accreted), “*parameters such as photolysis/radiolysis of HMT on meteorite parent bodies*” is negligible.

You may know that Tagish lake has 99 % IOM (overall) so the abundance of SOM is completely different than in Murchison and Murray. Even if this doesn't explain at all your results, it argues for big differences in the OM accreted.

[Reply] We added the following sentence “(e.g. HMT abundance when each parent body formed by accretion)”.

[Comment 18] *Line 186: please check your ref again, please add 23, and 27.

[Reply] Added.

[Comment 19] *line 189: ‘*both molecules are likely to be lost from grains during warming up phases*’. This is partially wrong; it is only if you considered that they will not react with others molecules. So I would suggest to add “if they do not chemically react”. As you know, many chemical reactions happened during the warming up phase, the older statement that it was only for desorption is obsolete. (<https://www.sciencedirect.com/science/article/abs/pii/S0273117713004183>)

[Reply] We added the following sentence “*unless transformed into other (non-volatile) species by chemical reactions*” here.

[Comment 20] *[Comment 37] -line 176 [if the temperature of the formed asteroids] this part of the sentence is not a the right place, since the process

you described happen before asteroids accretion. Please remove or rephrase.

[Reply] This statement is not necessarily true. There are some astronomical objects in the outer region of our solar system where the temperature is ~50 K or below, e.g. TNOs. In addition, as pointed out by other two Reviewers, a potential snowline drift may cause variations in the temperature of such objects, which may lead to the desorption of both molecules depending on the temperature. We do not limit our discussion only to the asteroid belt at ~2-3 AU from the Sun, and would like to be inclusive of environments across the entire solar system. What you are replying is not related to my comment. I am not discussing the temperature.

Your arguments are entirely true, but it was not my comment. ;) Please carefully read again your sentence, line 190. You have written: *“both molecules are likely to be lost from grains during warming up phases toward star formation if the temperature of the formed asteroid exceeds the desorption temperature of both molecules”* You are describing warming up phases during star formation, and then you jump to already accreted asteroids. They are two different episodes, no ? first grains are formed in molecular clouds, mixed, irradiated, warmed up (and for sur to much warmed in the inner solar region) before incorporation into asteroids or comets. Your sentence imply that your grains were already accreted into asteroids. I hope to be clearer this time for this comment. Can you rephrase to avoid the confusion please.

[Reply] In the previous review comment, we did not catch what Reviewer 2 meant. But this time it is clear. Then, the words “formed asteroid” are replaced with “**grains**”.

[Comment 21] *Line 195 “however, they have been “, replace they by ammonia and formaldehyde, when reading we don’t know anymore that it refers to ammonia and formaldehyde.

[Reply] Replaced.

[Comment 22] *line 199: I think you forget one ref, the 19, so ref should be 15-19.

Replies: NCOMMS-20-26492B

Thank you for the rest of your reply.

V. Vinogradoff

[Reply] Modified.